# AI Research Agents for Machine Learning: Search, Exploration, and Generalization in MLE-bench

**Edan Toledo**[*,1,2] **Karen Hambardzumyan**[*,1,2] **Martin Josifoski**[*,1] **Rishi Hazra**[†,1,3]
**Nicolas Baldwin**[1] **Alexis Audran-Reiss**[1] **Michael Kuchnik**[1] **Despoina Magka**[1] **Minqi Jiang**[1]
**Alisia Maria Lupidi**[1] **Andrei Lupu**[1] **Roberta Raileanu**[1] **Tatiana Shavrina**[1] **Kelvin Niu**[1]
**Jean-Christophe Gagnon-Audet**[1] **Michael Shvartsman**[1] **Shagun Sodhani**[1]
**Alexander H. Miller**[1] **Abhishek Charnalia**[1] **Derek Dunfield**[1] **Carole-Jean Wu**[1]
**Pontus Stenetorp**[2] **Nicola Cancedda**[1] **Jakob Nicolaus Foerster**[1] **Yoram Bachrach**[1]

[*]Equal contribution [†]Work done while at Meta

[1]Meta [2]University College London [3]Örebro University

## Abstract

AI research agents are demonstrating great potential to accelerate scientific progress by automating the design, implementation, and training of machine learning models. We focus on methods for improving agents' performance on MLE-bench, a challenging benchmark where agents compete in Kaggle competitions to solve real-world machine learning problems. We formalize AI research agents as search policies that navigate a space of candidate solutions, iteratively modifying them using operators. By designing and systematically varying different operator sets and search policies (Greedy, MCTS, Evolutionary), we show that their interplay is critical for achieving high performance. Our best pairing of search strategy and operator set achieves a state-of-the-art result on MLE-bench lite, increasing the success rate of achieving a Kaggle medal from 39.6 % to 47.7 %. Our investigation underscores the importance of jointly considering the search strategy, operator design, and evaluation methodology in advancing automated machine learning.

## 1 Introduction

Science is based on *searching* the open-ended space of hypotheses and *testing* them in a *controlled experiment* [27]. Recent breakthroughs have resulted in artificial intelligence (AI) agents that offer great potential to automate the scientific discovery process [49, 42, 3]. A key obstacle to improving research agents is that their designs entangle several factors for performance, making it difficult to pinpoint sources of improvement via controlled experiments at scale (Fig. 7). These factors span *algorithm design*, *concrete implementation*, and *ability to leverage compute*, as performance gains accrue only if no layer in the stack bottlenecks the benefits from additional compute resources. This challenge is

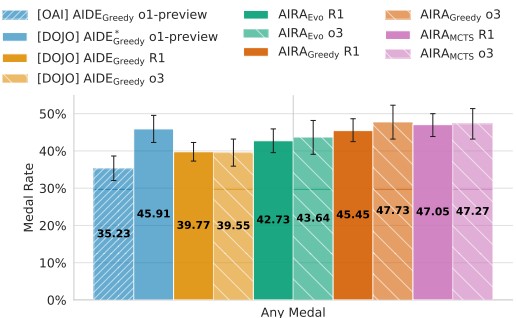

Figure 1: AIRA agents use the AIRA-dojo environment, AIRA operators, and search policies to achieve SOTA performance on MLE-Bench lite.

exemplified in MLE-bench [4], a benchmark where AI agents compete in Kaggle competitions to solve real-world machine learning (ML) problems. Notably, the state-of-the-art approach AIDE [23]

39th Conference on Neural Information Processing Systems (NeurIPS 2025).

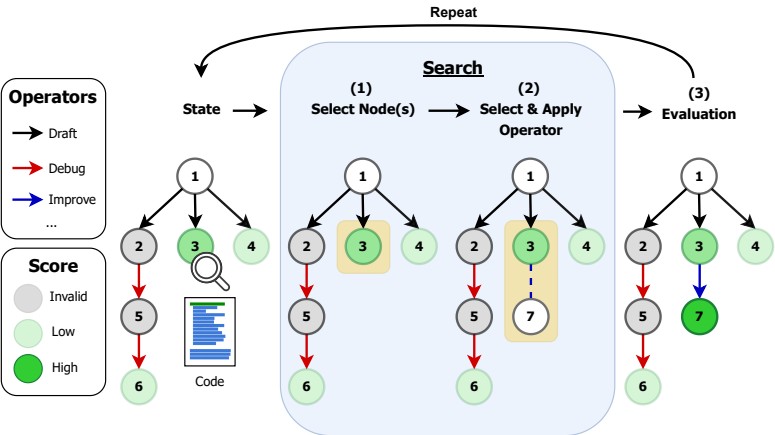

Figure 2: **Overview of AIRA.** Given a problem specification, AIRA maintains a search graph whose nodes are (partial) solutions. At each iteration, the agent (1) selects nodes via a *selection policy*, (2) picks an operator via an *operator policy* and applies this operator to the node, and (3) scores the resulting solution via a *fitness function*. Here, a greedy node-selection strategy applies the improve operator to the highest-scoring node.

does not fully disentangle these performance factors, providing limited insight into which components primarily drive the agent's performance and where improvements are needed.

**We start by formalizing the design of AI research agents as a search algorithm composed of two components**: The first one is the *search policy* which is used to navigate the space of candidate solutions, and the second is the *operators* which iteratively modifies existing solutions to generate new candidate solutions. In this framework (Fig. 2), AIDE is represented as a *greedy search algorithm* that, at each step, applies one of its code operators (i.e., DRAFT, DEBUG, IMPROVE) to the current best solution. This allows us to disentangle the effect of the *search* from that of the *operators*.

To assess alternative search algorithms, **we develop more sophisticated agents performing Evolutionary and Monte Carlo Tree Search** (MCTS) [6, 24]. We empirically show that AIDE's operators, rather than the search algorithm, are a bottleneck to better performance.

Based on these findings, **we design an improved set of operators** and evaluate them when paired with the above-mentioned search algorithms. Our best-performing agent achieves state-of-the-art results on MLE-bench lite, increasing the rate of achieving a Kaggle medal from 39.6 % to 47.7 %.

Additionally, **we investigate how the generalization gap—the difference between validation and test scores—affects the agents' performance**. In ML engineering tasks, as in many real-world use cases, an agent can access only proxy metrics (e.g. validation loss) to guide the search process, while the final solution is evaluated on a held-out test set. Therefore, a gap between the expected (validation) and actual (test) loss, could potentially mislead the search process. Indeed, we find systematic overfitting: selecting the final solution in a search graph by its test—rather than validation—score, would increase the medal rate by 9 to 13 % (absolute scale), depending on the search algorithm. These findings highlight the importance of rigorous evaluation protocols during search and regularization for robust and scalable scientific discovery.

Finally, to conduct experiments, **we develop AI Research Agent dojo (AIRA-dojo), a framework that provides a scalable and customizable environment for AI research agents**. First, AIRA-dojo exposes a robust and flexible interface to compute resources, which is essential for building effective agents. The baseline, AIDE, implemented in AIRA-dojo achieves a performance increase of 10.68 % (absolute scale) over the reported results [4]. Second, AIRA-dojo enables users to experiment with custom operators, search policies, evaluation methods, and tasks within a comparable setup. This facilitates a rigorous scientific study of AI research automation. Our code is open-sourced at: https://github.com/facebookresearch/aira-dojo.

## 2 Research Agents as Search Algorithms

AI research agents typically approach machine learning problems by generating artifacts – codebases designed to solve a given task. Executing these artifacts yields trained models, which are then evaluated against a chosen performance metric. The agent iteratively refines its artifacts to improve performance on this metric.

Previous research indicates that LLMs alone are insufficient to effectively solve such open-ended tasks [32]. In particular, LLM performance significantly improves when augmented with external tools [37], execution feedback [13], and solutions addressing context limitations. Therefore, developing high-quality models typically involves iterative experimentation, where insights from prior experiments inform subsequent refinements. Recent advancements by Jiang et al. [23] demonstrate state-of-the-art performance by conceptualizing this iterative experimentation as a tree search over potential solutions.

In this section, we formalize and generalize this perspective by modeling research agents as graph-based search algorithms. The proposed framework allows systematic exploration of alternative agent design choices, providing insights into how different algorithms affect the exploration-exploitation trade-off—a fundamental aspect of search [24, 44].

### 2.1 Graph–based Search Framework

We consider an agent that operates by searching a directed graph $\mathcal{G}_t = (V_t, E_t)$ that evolves over multiple iterations $t = 0, 1, \ldots$, where each node $v \in V_t \subseteq \mathcal{S}$ represents an artifact belonging to the set of all possible artifacts $\mathcal{S}$, while each directed edge $(v_i, v_j) \in E_t$ represents a transformation from $v_i$ to $v_j$. The root $v_0$ is the initial artifact that can represent an empty or starting artifact.

**Definition 1 (Components of the Search Algorithm).** A graph-based search algorithm is specified by the tuple $(\mathcal{F}, \pi_{\text{sel}}, \mathcal{O}, \pi_{\text{op}}, \tau)$:

- **Fitness Function.** $\mathcal{F} : \mathcal{S} \to [0; 1]$ is a function that estimates the value or quality of a node $v \in V_t$. Since true fitness is typically not available, $\mathcal{F}$ is often a proxy measure of the value of the node.

- **Selection Policy.** $\pi_{\text{sel}} : 2^{V_t} \to 2^{V_t}$ chooses a subset of nodes $U_t \subseteq V_t$ on which to operate, typically guided by a heuristic function $h : V_t \to \mathbb{R}$, which assigns a scalar estimate to each node. The heuristic may be derived directly from the fitness function $\mathcal{F}$, such as the upper confidence bounds for trees (UCT) used in MCTS [24], or other custom heuristics tailored to the domain [5, 18].

- **Operator Set.** $\mathcal{O} = \{ o_\ell : 2^{\mathcal{S}} \to \mathcal{S} \}_{\ell=1}^{L}$ comprises $L$ transformation functions that propose new artifacts $v = o_\ell(\{v_k\}_{k=1}^m) \in \mathcal{S}$ from one or more selected artifacts. In AIDE, examples include DRAFT, DEBUG, and IMPROVE instantiated as prompt-based instructions to an LLM. Composite operators (e.g. the result of applying a sequence of base operators) can also be used.

- **Operator Policy.** $\pi_{\text{op}} : \mathcal{O} \times U_t \to \mathcal{O}$ decides which operator to apply to the current node selection.
- **Termination Rule.** $\tau$ halts the search when a computational budget is exhausted, progress stalls, or a fitness threshold is reached.

At each iteration, the agent selects existing promising artifacts, applies transformation operators, and updates the search graph, propelling the discovery process forward. We discuss specific instantiations of search algorithms in Sections 2.3, 4.2. In all our instantiations: ① the termination criterion is set as the **wall-clock** time or the **maximum number of artifacts**, whichever happens first; ② the fitness function $\mathcal{F}$ is defined for each node by the operator that generates or modifies it (i.e., $\mathcal{F}$ **is not global**); ③ all operators are LLM-driven except for the MEMORY operator, which is defined by hand.

### 2.2 Operators

We define operators as high-level functions that take in existing artifacts and produce new ones. These can range from simple rule-based parsers to LLM calls using prompting techniques [46, 53] or more complex agents like Cursor [20].

This broad definition enables a unified comparison across search methods. A search algorithm can be instantiated with any mix of LLM-based, tool-based, or nested search operators.

Building on the demonstrated effectiveness of AIDE [23], we adopt a similar operator set as introduced in their framework, consisting of the following: ① DRAFT initializes the search process by generating

an initial population of candidate artifacts. ② DEBUG attempts to identify and correct errors in invalid artifacts. ③ IMPROVE refines valid artifacts to enhance their performance according to the evaluation criteria. ④ MEMORY chooses how and where information from past artifacts is used in subsequent operations. ⑤ Furthermore, we introduce CROSSOVER which recombines useful elements from two artifacts to create a new candidate. Section 3 contains further details.

## 2.3 Re-casting AIDE in our Notation

AIDE [23] is an LLM-driven agent that frames problem-solving as a tree-search over Python scripts. In the tuple $(\mathcal{F}, \pi_{\text{sel}}, \mathcal{O}, \pi_{\text{op}}, \tau)$ from Section 2.1 its components are:

**Fitness & selection ($\mathcal{F}, \pi_{\text{sel}}$).** For each node $v$, fitness is the mean 5-fold cross-validation (CV) score $\mathcal{F}(v) \in [0, 1]$[1]. The selection policy is greedy with respect to $\mathcal{F}$. This means at iteration $t$ the agent selects and operates on $v^\star = \arg\max_{v \in V_t} \mathcal{F}(v)$ but, with probability $\varepsilon_{\text{bug}}$[2], may instead revisit a *buggy* node ($\mathcal{F} = 0$) to aid recovery and maintain diversity.

**Operator set $\mathcal{O}_{\textbf{AIDE}}$.** The agent exposes three LLM operators {DRAFT, IMPROVE, DEBUG} and one handcrafted operator MEMORY or as defined by Jiang et al. [23]—the SUMMARIZATION operator. The operators are designed to output the following: DRAFT (3–5-sentence plan + fenced script that trains, evaluates, and writes `submission.csv`); DEBUG (short diagnosis and repaired script given a traceback); IMPROVE (Creating exactly one measurable change — feature, architecture, schedule, etc. — in plan + code form); MEMORY (running summary of all previous designs, scores, and notes that is appended to every DRAFT/IMPROVE prompt).

**Operator policy $\pi_{\textbf{op}}$.** (a) *Seeding:* invoke DRAFT exactly $n_d$ times from the root $v_0$; (b) *Logging:* after every DRAFT or IMPROVE, call MEMORY; (c) *Main loop:* for the node chosen by $\pi_{\text{sel}}$ apply IMPROVE if it is valid and at least $n_d$ drafts exist, if less than $n_d$ draft nodes exist, DRAFT is called, otherwise apply DEBUG.

This combination of $\pi_{\text{sel}}, \pi_{\text{op}},$ and $\mathcal{O}_{\text{AIDE}}$ defines the baseline agent we denote AIDEGREEDY (also summarized in Section B). When the agent uses an alternative search policy while keeping $\mathcal{O}_{\text{AIDE}}$ unchanged, we write AIDE$_{\text{search\_policy}}$ to identify change. For example, AIDEMCTS uses the AIDE operator set and MCTS as the search policy.

# 3 Experiment Design

## 3.1 AIRA-dojo

An agent's environment greatly influences its performance. To systematically study the space of agentic policies (see Fig. 7), we introduce the AI Research Agent (AIRA) dojo—a scalable and customizable framework for AI research agents. AIRA-dojo provides the *environment* in which agents operate, along with abstractions for *operators* and *policies* as described in Section 2, and *tasks* that define evaluation criteria for agent performance. Using these abstractions, we implement and evaluate four search policies: MCTS, Evolutionary, Greedy, and our own implementation of AIDE. We hope that AIRA-dojo's scalability and customizability, together with the provided agent and task implementations, will support and advance future research in the community.

The infrastructure design of AIRA-dojo was informed by several reliability and performance constraints, as discussed in more detail in Section G.

## 3.2 Environment

The environment defines the context in which agents operate.

**Action Space.** We use Jupyter notebooks to execute code from the agents. With this interface, agents can execute arbitrary Python code, perform file reads and writes, and even use the shell via Jupyter magic commands. The environment captures and returns the status, standard outputs, tracebacks for debugging, and the code block execution time to the agents.

---

[1]In practice, CV scores are not necessarily bounded between zero and one; for example, RMSE is an unbounded metric.

[2]$\varepsilon_{\text{bug}}$ is set to 1.0 so as to mimic the hyperparameters used in MLE-bench by Chan et al. [4].

**Isolation.** The environment enforces program-level constraints, such as total runtime limit, GPU and CPU usage, memory, and storage, using `Apptainer` containers [26]. This ensures complete isolation from host systems, preventing agents from affecting the host environment or other agents, and mitigates risks from unintended actions or failures, such as data leakage or interference with other processes. Furthermore, agents possess root-like privileges within their isolated containers, granting them full control over their environment. This allows them not only to configure their environments using standard package management tools such as `pip`, `conda`, but even `apt-get install`.

**Superimage.** A container image provides a base set of tools required for ML tasks. This image includes pre-configured CUDA support, deep learning frameworks like `PyTorch` and `TensorFlow`, and essential data science libraries. Each coding session starts from the original `Superimage` state and can diverge based on the agent's actions while isolating the state of the other agents.

Together, these design choices create a robust testbed that eliminates confounders at both the system and implementation levels. This enables reproducible benchmarking and facilitates the development of long-running agentic systems across thousands of parallel runs.

### 3.3 MLE-bench

MLE-bench [4] contains 75 Kaggle-sourced tasks for evaluating machine learning engineering agents. We evaluate on MLE-bench lite – a curated subset of 22 tasks selected from the full benchmark – allowing us to allocate more seeds per task and increase our confidence in our results.

Our experiments showed that existing methods exhibit high variance in this benchmark (see Section J). In line with the benchmark guidelines, we assess each agent's performance using the **Medal Success Rate**. Specifically, for each task, agents earn a bronze, silver, or gold medal according to task-specific percentile thresholds. We report the percentage of attempts in which an agent secures a medal.

### 3.4 Experimental Details

**Environment.** Each candidate agent is launched in a freshly initialized, sandboxed process in AIRA-dojo. This guarantees that file systems and environment variables are isolated across evaluations, and there is no cross-agent interference. Every sandbox is provisioned with a fixed hardware quota: 1 dedicated H200 GPU, 24 logical CPU cores, 100 GB of RAM, and 1 TB of additional scratch storage. Internet access is permitted solely for fetching third-party packages and model checkpoints.

**Time Constraints.** In line with MLE-bench, the agent is allowed a 24-hour wall-clock window. Within this period, each agent has a maximum runtime of 4 hours per code execution. We chose to reduce this from the 9-hour limit used in MLE-bench after preliminary experiments showed no difference in performance and a higher average number of valid nodes in the search trees.

**LLMs.** We conducted all the experiments with the full-sized `DeepSeek R1` [7] model, with 128K-token context window to ensure input coverage without truncation. Due to wall-clock constraints, this choice was guided by both the model's capabilities and applicable rate limits. In particular, we selected `DeepSeek R1` as the most capable open-source model available, which allowed us to self-host inference servers and maintain high experimental throughput without encountering rate limits. For the main results, we also evaluated `o3` [35], one of the most capable closed-source models. To ensure experiment validity and avoid hitting rate limits, we limited the number of parallel runs when experimenting with `o3`. We always use GPT-4o [34] with Structured Outputs to parse code execution outputs—extracting run success, summary text, and validation metric. For the figures, we take `[OAI] AIDE o1` artifacts from the MLE-bench [4] GitHub repository. To directly compare with their results, we evaluated `o1-preview`, but only completed experiments for one agent before deprecation. See Section E for details.

## 4 AIRA

### 4.1 Operators $\mathcal{O}_{\mathbf{AIRA}}$

As part of AIRA-dojo, we propose a new operator set based on $\mathcal{O}_{\mathrm{AIDE}}$, denoted $\mathcal{O}_{\mathrm{AIRA}}$. To this end, we focus on maintaining a cleaner context through better-scoped memory, and encouraging structured reasoning and strategic diversity in ideation. The key differences are:

① **Prompt-adaptive complexity.** We introduce a dynamic complexity cue within the system prompt in order to guide the complexity of artifacts generated by the DRAFT and IMPROVE operators. The complexity is determined by the number of children, $n_c$, of the node being processed:

complexity$(n_c)$ = "minimal" if $n_c < 2$, "moderate" if $2 \leq n_c < 4$, "advanced" if $n_c \geq 5$

For the DRAFT operator, this cue influences the complexity of the generated ideas. For the IMPROVE operator, it guides the complexity of the enhancements. This dynamic signal helps prevent premature over-engineering by ensuring the agent provides simple solutions when appropriate, while encouraging more thorough exploration when more advanced solutions may be necessary.

② **Scoped memory.** We modify the MEMORY operator to extract different types of memories depending on the operator used. Specifically, for DRAFT and IMPROVE, it retrieves only *sibling memories*—the children of the artifact the agent is applying the operator to—thereby promoting diversity. This prevents overloading the context and reduces behavior indicative of mode collapse. Conversely, for DEBUG, it retrieves the entire *ancestral memory* of the artifact's debug chain, enabling review of prior fix attempts and avoiding "undo–redo" oscillations.

③ **Think Tokens.** For reasoning models, we use the operators' system prompts to explicitly encourage them to use thinking tokens for reasoning and reflection. These thoughts are stripped from the final answer–remaining invisible to other operators (e.g., memory). On average, we observe a $2\times$ increase in completion tokens generated by the AIRA operator set (see Section F).

### 4.2 Agents

In this section, we introduce three agents that combine the operators $\mathcal{O}_{\text{AIRA}}$—proposed in Section 4.1—with distinct search policies (see Appendix H for the rationale behind their selection). Each agent uses the same proxy–fitness function $\mathcal{F}$ (5-fold CV) and the same termination criterion $\tau$ (based on wall-clock time or artifact cap).

**AIRAGREEDY**. This agent employs greedy search (Section 2.3) using the $\mathcal{O}_{\text{AIRA}}$ operator set. Any performance improvement of AIRAGREEDY over AIDEGREEDY directly reflects the benefit of the new operators, since the *only* difference between the two is the operator set.

**AIRAMCTS**. This agent uses Monte-Carlo Tree Search (MCTS) [24] with $\mathcal{O}_{\text{AIRA}}$ operator set. Our implementation of MCTS follows the canonical loop (selection, expansion, evaluation, and backup), but omits simulated roll-outs: the leaf value of expanded nodes is the proxy fitness function $\mathcal{F}$:

- *Selection.* From the root node $v_0$, descend by selecting the node with the highest UCT score $\pi_{\text{sel}}(v) = \arg\max_{v \in V_t} h_{\text{UCT}}(v)$ where $h_{\text{UCT}}(v \mid u) = Q(v) + c\sqrt{\log N(u)/(N(v) + \varepsilon)}$, where $N(\cdot)$ and $Q(\cdot)$ are the visit count and running mean fitness. Here, $u$ is the parent of $v$.
- *Expansion.* Only leaves are expanded. The chosen operator from $\mathcal{O}_{\text{AIRA}}$ is applied $n$ times, creating $n$ children. Buggy children enter an automatic DEBUG loop until fixed or the budget expires.
- *Evaluation & backup.* The leaf fitness is $\mathcal{F}(v_\ell)$ is back-propagated to ancestors with the standard incremental update of $(N, Q)$.

**AIRAEVO**. The evolutionary agent keeps a population $V_t$ of fixed size $n$ and repeats:

- *Parent selection:* Select individuals with probability $\pi_{\text{sel}}(v) = \mathcal{F}(v)/\sum_{u \in V_t} \mathcal{F}(u)$.
- *Reproduction:* With a fixed probability, apply IMPROVE; otherwise, apply CROSSOVER. Buggy parents undergo DEBUG until they are either fixed or the debug attempt limit is reached.
- *Replacement:* Offspring replace the least-fit individuals in $V_t$.

For both AIRAMCTS and AIRAEVO, we normalize all fitness values using the minimum and maximum values observed during the search process. This normalization ensures a consistent set of hyperparameters throughout the search. To select the final solution, the AIRA and AIDE agents return the one with the highest validation score.

## 5 Experiments and Results

### 5.1 Analyzing the Performance of the Current SoTA

The most effective search algorithms strike a balance between exploration and exploitation. In this section, we analyze the exploration–exploitation trade-offs of AIDE, the current state-of-the-art method, and then investigate how additional computation time increases its performance.

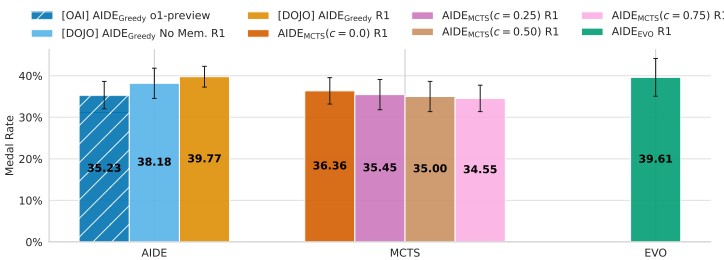

Figure 3: **Searching with AIDE's operators.** When limited to AIDE's operator set $\mathcal{O}_{\text{AIDE}}$, agents using more advances search policies (e.g., MCTS, evolutionary algorithms) gain no advantage, underscoring the operator set as the bottleneck.

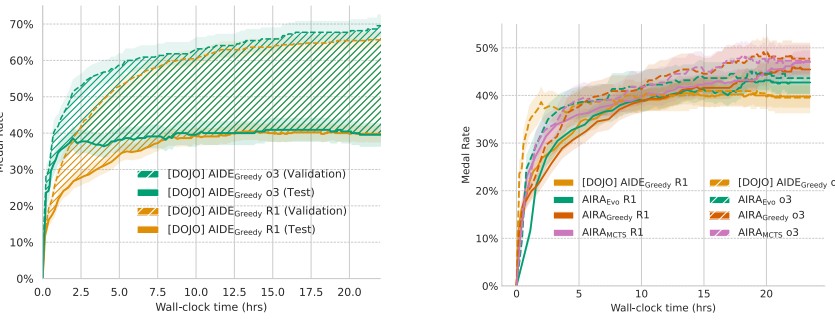

Figure 4: **a) AIDE's performance profile over 24-hour search window.** Perceived vs. actual medal rate over 24 hours of AIDEGREEDY. The curves show the mean validation (agent-reported) and held-out test medal rates across 20 seeds for all tasks. The widening band illustrates the generalization gap, revealing how apparent gains on the validation set can mask overfitting and ultimately undermine the search process. **b) Performance profiles of all agents after 24-hour search window.**

The three main factors that influence this balance are memory, the operator set, and the selection policy. **Memory** structures prior knowledge—storing promising solutions and tracking which regions of the solution space have been sampled—to inform subsequent decisions. This information can bias the search toward exploration by encouraging diversity or toward exploitation by discouraging it. **Operators** then apply controlled transformations to existing solutions: for instance, random mutations introduce diversity, targeted refinements exploit known strengths, and recombinations merge features from multiple candidates. Finally, the **selection policy** balances exploiting high-quality regions of the solution space with exploring less-tested areas by determining how to allocate computational resources. A non-greedy selection policy, such as in MCTS, periodically directs resources toward branches that may appear suboptimal, with the goal of uncovering better solutions.

**What is the effect of memory?** To quantify the impact of the MEMORY operator, we conduct a controlled ablation comparing the performance of AIDE with and without the MEMORY operator enabled. As shown in Fig. 3, both variants achieve nearly identical mean medal rates. This suggests that memory is not a driving factor behind AIDE's strong performance. **What is the effect of exploration at the search level with AIDE's operators?** AIDE uses a selection policy that does not explore and always greedily selects the most promising candidate. To evaluate the impact of the search policy as a modulator of the exploration-exploitation tradeoff when searching using AIDE's operators, we replaced the operators in AIRAMCTS with that of AIDE and varied the UCT exploration constant $c_{\text{UCT}} \in \{0, 0.25, 0.75\}$. The $c_{\text{UCT}}$, which controls exploration in MCTS, allows us to isolate the effect of search-level exploration on downstream performance. The resulting medal rates (Fig. 3) showed only marginal differences across all $c_{\text{UCT}}$ values, supporting our hypothesis that search-level exploration is constrained by the interaction between the operator set and the search policy. We observe the same limitation when performing the same operator replacement in AIRAEVO.

**What is the performance profile of AIDE?** To examine the improvement gains over time, we plot AIDE's anytime performance, which is the average medal rate achieved by the agent if the search was

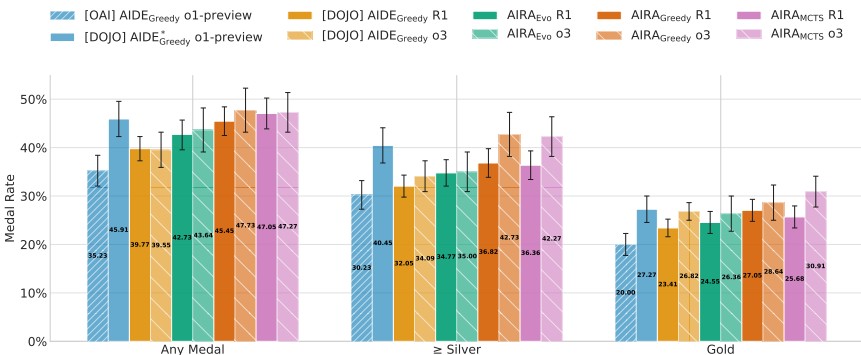

Figure 5: **Medal rates on MLE-bench Lite.** Performance is shown for three medal categories: any medal, silver medals and above, and gold medals only. Error bars represent 95% confidence intervals computed using stratified bootstrapping.

to terminate at time $t$. The results, summarized in Fig. 4a, suggest that while the agent's perceived performance (validation score) continues to improve over time, the true test performance plateaus, even slightly decreases over time. These findings suggest that overfitting might be a fundamental limitation to the agent's performance. We further investigate this in Section 5.3.

## 5.2 AIRA Beyond Greedy

Based on the observation that search policies yield no performance gain with AIDE's operators, this section is divided into two parts: ① assessing the effectiveness of our improved operators (see Section 4.1), ② examining their interplay with the more advanced search policies. The results are summarized in Fig. 5, with a detailed breakdown of performance for each task in Section L.

**Evaluating the environment.** But first, we highlight the benefits from AIRA-dojo. Specifically, our baseline agent implementation, AIDEGREEDY o1-preview, operating in the AIRA-dojo environment, improves the medal rate from 35.2 % to 45.9 % compared to the reported results [4]. This corresponds to state-of-the-art performance with a relative improvement of 30 % in medal rate. It is notable that AIDEGREEDY o1-preview, outperforms both AIDEGREEDY R1 and AIDEGREEDY o3, despite o3 being the newest model in the series. While evaluating all agents with o1-preview would have been informative, we were only able to complete the AIDEGREEDY experiment before the model was discontinued.

**Evaluating the operator sets.** AIDEGREEDY and AIRAGREEDY employ the same search policy but differ in their operator sets. Comparing their performance isolates the effect of the operators. AIRAGREEDY outperforms AIDEGREEDY (45.5 % vs. 39.8 %), representing a 14 % relative improvement and underscoring the importance of operators for performance.

**Evaluating search policies.** Equipped with an improved set of operators, we now explore whether combining them with advanced search methods can further enhance performance. The results (Fig. 5) show that for R1, AIRAMCTS achieves the best performance, achieving state-of-the-art performance on MLE-bench Lite with an average medal rate of 47%. Both AIRAMCTS and AIRAGREEDY achieve silver-or-above medal rates ($\approx 36.5$%), outperforming the baseline by 4.5 absolute points. In terms of gold medals, AIRAGREEDY performs best, reaching 27%. For o3, AIRAMCTS and AIRAGREEDY perform comparably in terms of any medal and silver-or-above medal rates, while AIRAMCTS outperforms AIRAGREEDY in gold medals by 2.2 percentage points. Although agents using R1 and o3 achieve similar overall medal rates, those using o3 perform better when considering gold and silver-or-above medal rates. Specifically, AIRAMCTS increases the gold medal rate from 25.68 % with R1 to 30.91 % with o3, a relative increase of 20 %.

Collectively, what stands out is that all search policies combined with AIRA operators outperform AIDEGREEDY. Furthermore, the rankings between agents using the AIRA operators with different search strategies differs significantly from that observed in Section 5.1.

Finally, Fig. 4b shows the anytime performance of each agent over the 24-hour window defined by MLE-bench. For conciseness, we will focus our discussion on performance with R1, as the results

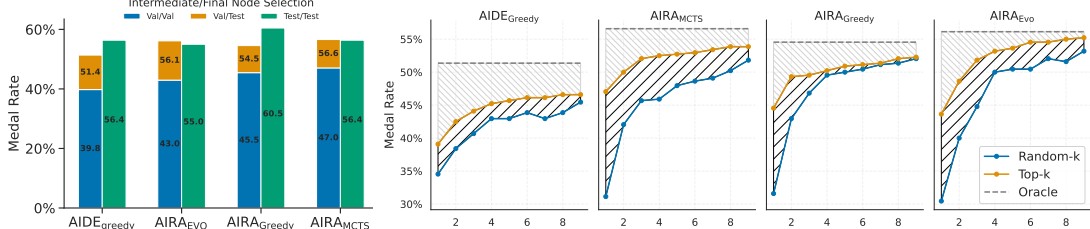

Figure 6: **a) The validation–test metric mismatch.** Results shown are for agents using R1. Bars depict the absolute performance gap between three configurations: (i) using the validation metric for both intermediate (search) and final (submission) node selection; (ii) using the validation metric only for search and the test metric for selection; and (iii) using the test metric for both search and selection. **b) Bridging the validation–test gap.** Medal rate achieved by two final node selection strategies as a function of $k$: (i) randomly sampling $k$ nodes and reporting the highest test score among them; (ii) selecting the top $k$ nodes by validation score and reporting the highest test score among those. As $k$ increases, the validation-based strategy closes the gap to the upper-bound performance given by the best test score over the entire search graph.

with o3 follow a similar pattern. We observe that the rankings between agents change over time. For example, at the 3-hour mark, the performance gap between AIRAGREEDY and AIRAMCTS is notable. This gap narrows by the 10-hour mark, where all agents perform similarly. Meaningful divergence in performance appears only after 19 hours. These trends reflect the interaction between rankings and the resources provided, as shown in Figure 7.

We further examine the effect of extended compute time on performance by running experiments for up to 5 days (Appendix C), showing that our agents continue to improve beyond the 24-hour period. To illustrate differences in search behavior, Appendix M presents representative search trees from our experiments for each agent, which may help clarify the strengths and weaknesses of each method.

### 5.3 The Generalization Gap: Searching with a Proxy Evaluation

Agents steer their search using the validation scores of candidate solutions, but ultimately, performance is measured on a held-out test set. Due to finite sample effects, the validation score is not perfectly predictive of performance on the test set—a discrepancy known as the *generalization gap*.

In this section, we measure the impact of this generalization gap on the agents' performance. We further quantify whether this impact is more detrimental during the intermediate node selection in the search process or the final (submitted) node selection.

**How large is the impact of the generalization gap on performance?** We begin by comparing two extremes: VAL/VAL—both intermediate and final node selection use the validation score (standard practice); TEST/TEST—an oracle baseline using the true test score at both stages. As shown in Fig. 6a, searching based on the test score instead of the validation score improves performance by 9.4 % and 12.4 % for AIRAMCTS and AIRAEVO. The gap is even higher for the agents using a greedy search policy, reaching 15 % for AIRAGREEDY and 16.6 % for AIDEGREEDY. For the rest of this section, we investigate the nature of this gap in performance and how to close it.

**How much of the gap can be attributed to final node selection?** To answer this question, we consider the following two settings: VAL/VAL; and VAL/TEST—the intermediate node selection uses the validation score, while the final node selection is made based on the test scores. The results (Fig. 6a) indicate that by selecting the best node in a search graph constructed without privileged access, i.e., based on validation signal, all agents achieve a performance boost of 9 to 13 absolute points. Crucially, comparing VAL/TEST with TEST/TEST shows that oracle final-node selection eliminates the gap between the standard (VAL/VAL) and oracle (TEST/TEST) settings for both AIRAMCTS and AIRAEVO. Even for the agents implementing the simpler greedy policy, selecting the best node in the final search graph closes more than 60 % of the gap. These findings highlight robust final-node-selection strategies as a promising path to higher performance.

**Bridging the gap through multiple submissions.** A straightforward remedy is to hedge against validation score noise by selecting multiple promising nodes. Specifically, among the top-$k$ nodes

ranked by validation score in the search graph, we report the test score of the best-performing one. As a baseline, we repeat the process for a set of random-$k$ nodes, and summarize the results in Fig. 6b. We highlight the following observations: ① the larger gap between top-k and random-k in agents using non-greedy algorithms, AIRAEVO and AIRAMCTS, suggest greater diversity in their search graphs; ② with as little as 3 top-k submissions agents achieve an additional 10 % of performance; ③ the top validation nodes are informative of the best performing nodes.

## 6 Related Works

**Scaling Search with LLMs.** Integrating search with LLM-based generators has gained traction across domains such as coding [1], math [52], and planning [16, 15]. In practice, increased test-time compute through algorithms like best-of-$N$[45], beam search [16], MCTS [1, 15], and evolutionary search [39, 17] often improves performance beyond the architectural and size constraints of LLMs [41]. Our results highlight that these performance gains depend critically on the interplay between search components (see Section A)—an aspect that prior work has largely overlooked.

**Automating ML Engineering and Scientific Discovery.** Traditional approaches like AutoML [12, 43, 33] and Neural Architecture Search (NAS)[54, 29, 38] automate ML by searching over predefined, expert-designed configuration spaces, often via brute-force or heuristic methods [2, 8, 28]. In contrast, recent advances in LLMs enable more open-ended design. AIDE [23] treats ML engineering as LLM-guided code optimization via tree search. AI Scientist v2 [49] extends this paradigm, automating the entire research pipeline using agentic LLMs. Similar techniques have been applied to software engineering [1], algorithm and reward design [39, 31, 11, 17, 7], and even natural sciences [3, 42]. Existing systems often combine search procedures, operators, and evaluation in ways that make it challenging to understand which components drive performance improvements. To address this, we separate these components and look at how each one—and their interactions—can be optimized better. Concurrent work, R&D-Agent [51], which addresses the same problem setting, achieved impressive results on MLE-Bench. Our results fall within their reported standard deviation. We note that their experiments use at most 6 seeds, and as discussed in Section J, the limited number of seeds may introduce variation in the conclusions. Finally, our approach differs from methods such as Agent K [14], which uses long-term memory across multiple Kaggle competitions. In contrast, we focus on addressing each competition independently.

**AI Research Frameworks.** Most existing benchmarks for machine learning or research engineering, such as MLGym [32], RE-bench [48], and MLAgentBench [19], come with their own frameworks. Our approach is most similar to Inspect [21], a framework which provides an abstraction that makes minimal assumptions about the agent and evaluation design, clearly separating the two. This flexibility in agent design is essential for enabling rigorous comparisons across a wide range of agents—from simple LLM-based agents with tool access to scaffolds combining algorithmic and LLM-based components—within the same environment. The environment plays a critical role in performance (see Section 5.2). However, unlike Inspect, AIRA-dojo focuses strongly on long-running research tasks, which influences our environment design choices. For example, prior work typically uses Docker to containerize agents' workspaces, but Docker is unsupported on most HPC clusters due to its reliance on root privileges and lack of seamless integration with common HPC resource managers. This limitation hinders scalability, addressed through our Apptainer (see Section 3.2).

## 7 Conclusion

We conceptualize the design of AI research agents as a combination of two axes: search policy and operators. This formulation allowed us to perform a systematic investigation of the interplay between the two, highlighting how the operator set can act as a *bottleneck* to performance improvements. Based on this finding, we designed an enhanced operator set and constructed agents that combines these operators with several search strategies: Greedy, Monte Carlo Tree Search (MCTS), or Evolutionary Search. Our best-performing agent achieves a new state of the art on MLE-bench, increasing the success rate of winning a Kaggle medal from 39.6 % to 47.7 %. Further, we analyzed the role of the generalization gap in node evaluation. Our findings indicate *systematic overfitting*: selecting the final solution from the search graph based on its test score instead of its validation score would increase the medal rate by 9 to 13 % (absolute scale), depending on the search strategy. This highlights that robust final-node selection is a promising avenue for improving performance.

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

# Appendix

## A    Preconditions for Scaling

The hierarchical diagram in Fig. 7 illustrates key insights into the preconditions for improving agent performance through additional compute. Agents benefit from scaling search only when performance gains are not limited by the environment in which the agent operates, the quality of the evaluation signal guiding the search, the capability of the operators performing the search, or the search policy itself.

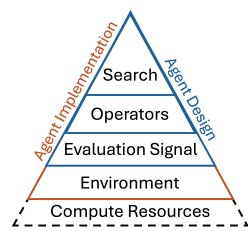

Specifically, if the agent is provided with $10\times$ more compute resources (e.g., 10 GPUs for 24 hours), the environment must allow the agent to effectively leverage these resources. Without a sufficiently high-quality evaluation signal—the ability to accurately assess solution quality—the

Figure 7: **Key factors dictating agent performance and scalability as compute resources increase.**

direction of improvement will be unclear, which would undermine the search process. Finally, a successful search requires capable operators that effectively perform their functions and a search policy that allocates compute efficiently to the most promising regions of the search space and appropriate operators.

Overall, this work underscores the importance of a strong foundation in algorithm design and infrastructure to ensure that scaling search translates into downstream performance improvements.

## B    Re-Casting AIDE in our Notation

Section 2.3 shows how AIDE can be reformulated within our framework. Table 1 summarizes the tuple that specifies the AIDE agent.

Table 1: Instantiation of the tuple $\big(\mathcal{F}, \pi_{\mathrm{sel}}, \mathcal{O}, \pi_{\mathrm{op}}, \tau\big)$ for AIDE.

| Component | AIDE choice |
|---|---|
| $\mathcal{F}$ | 5-fold CV score |
| $\pi_{\mathrm{sel}}$ | Greedy + $\varepsilon_{\mathrm{bug}}$ exploration |
| $\mathcal{O}$ | $\mathcal{O}_{\mathrm{AIDE}}$ |
| $\pi_{\mathrm{op}}$ | Fixed rule |
| $\tau$ | Wall-clock or No. Node cap |

## C    The Effect of Compute: Searching Beyond 24h

In our analysis of the agents' anytime performance (see Section 5.2), we observe that their rankings evolve over time, with significant differences in performance emerging only after 15 hours of execution. For the experiments presented in the main paper, we adopt the experimental setup proposed in the MLE-bench paper, where agents are given 24 hours to complete the task. However, our aim is to develop agents capable of effectively utilizing computational resources well beyond the 24-hour limit, and thus, evaluating within this restricted timeframe offers only limited insights into their potential long-term capabilities.

To examine the impact of access to computational resources, we conducted an experiment with the same setup described in Section 5.2, extending the total runtime to 90 hours—nearly four days. The results, summarized in Fig. 8, demonstrate a similar pattern, where notable differences in behavior emerge after roughly 15 hours. Specifically, the performance of the AIRA agents (AIRAGREEDY and AIRAMCTS) continues to improve, whereas the performance of AIDEGREEDY plateaus. Another notable observation is that agents employing our proposed operators, AIRAGREEDY and AIRAMCTS, exhibit comparable improvement profiles until around the 50-hour mark, at which point AIRAMCTS

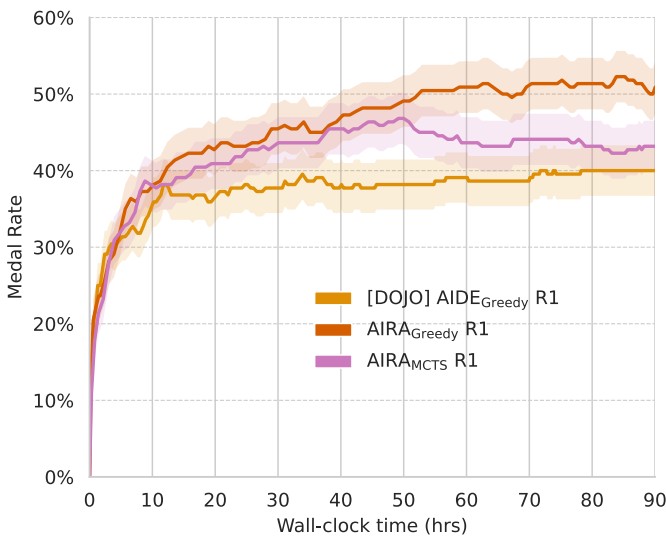

Figure 8: **Medal achievement rates over a 90-hour search horizon.** Each point represents the mean percentage of medals earned across 10 independent runs per task on the MLE-bench lite suite. Results are based on a complete replication of the baseline experiments and extended to 90 hours of search.

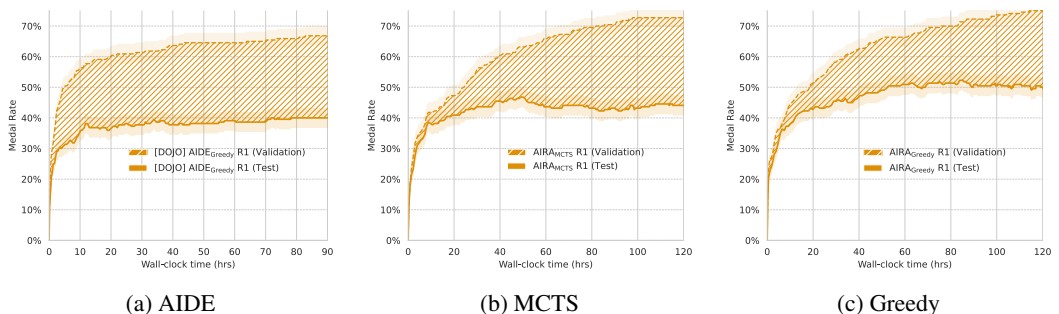

| (a) AIDE | (b) MCTS | (c) Greedy |

Figure 9: **Test vs. Validation Medal Rates over 90 hours.** Comparison of medal rates between test and validation scores for up to 120 hours of runtime. Each subplot shows the medal rate if the validation score matched the test score (potential performance) versus the actual medal rate given the true test score, revealing the performance gap between validation and test performance.

begins to overfit and its performance subsequently deteriorates. Conversely, AIRAGREEDY continues to improve, ultimately achieving a peak medal rate of approximately 53%. This peak represents an absolute improvement of 6% over the highest performance attained with 24 hours of compute, translating into a relative gain of 12%. However, the agent's performance subsequently declines as overfitting begins to manifest. We see in Fig. 9 that the validation performance continues to climb higher and overfitting becomes more severe. Ultimately over the longer search horizon, we see AIRAGREEDY and AIRAMCTS achieve a similar test-validation gap as AIDEGREEDY. This is consistent with our findings in Section 5.3 and highlights a fundamental limitation that affects agents' ability to operate effectively over much longer time horizons.fu

Overall, this analysis underscores the necessity of jointly considering search strategy, operator design, and computational resource availability (see Fig. 7). Furthermore, it highlights the fundamental impact of the generalization gap, identified and studied in Section 5.3.

# D   Evaluation on the Full MLE-bench Set

To get a better estimate of performance, as well as the effect of task difficulty and recent, in this section, we report the performance of agents on the full MLE-bench set of tasks. Specifically, we focus on our greedy agent, AIRAGREEDY, using o3 as the backbone LLM. We also report the results for AIDEGREEDY, the strongest version of the previous state-of-the-art agent, and for OpenAI's proprietary agents using o3 and gpt-5-thinking backbones, using their reported numbers. The average performance and the confidence intervals are computed over 20 random seeds.

Table 2 reports the performance for the overall performance, the complexity-based splits suggested by the MLE-bench paper [4], and the MLE-bench 30 split of interesting and diverse papers proposed in the gpt-5 system card [36]. We achieve an overall medal rate of 31.6%, and outperform the baseline on all splits. In all splits except from low, we achieve a relative increase of more than 100%. We also observed a 7.3 point performance increase on the low split (from 47.7 to 55.0) by using high reasoning effort and further improving the infrastructure (e.g., optimizing the file system and caching models used in solutions to avoid rate-limiting errors).

Table 2: Medal rates (mean ± standard error) across agents. The results annotated with an asterix are reproduced from prior work. For OpenAI o3 and GPT-5 Thinking, we report the values from the system card (`https://cdn.openai.com/gpt-5-system-card.pdf`). For AIDE (o1-preview), we use results from the MLE-Bench repository for all metrics, except for MLE-Bench-30, which we compute from trajectories on the MLE-Bench leaderboard (`https://github.com/openai/mle-bench/`).

| Model | All | Low | Medium | High | MLE-bench-30 |
|---|---|---|---|---|---|
| OpenAI agent, o3, no browsing | – | – | – | – | 6* |
| OpenAI agent, gpt-5-thinking, no browsing | – | – | – | – | 8* |
| AIDE Greedy (o1-preview) | $16.9 \pm 1.1$* | $34.3 \pm 2.4$* | $8.8 \pm 1.1$* | $10.0 \pm 1.9$* | $12.5 \pm 0.8$ |
| (Ours) AIRA Greedy o3 | $31.6 \pm 0.8$ | $55.0 \pm 1.5$ | $22.0 \pm 1.2$ | $21.7 \pm 1.1$ | $25.8 \pm 1.3$ |

To provide a comprehensive view of performance, we report the per-task medal rate in Table 3

# E   Implementation Details

Our experiments were run on a single-agent setup powered by an NVIDIA H200 GPU, 24 CPU cores of Intel(R) Xeon(R) Platinum 8488C, 100 GB of RAM, and an extra 1 TB of local-disk scratch space. Each job had a hard wall-clock cap of 24 hours, an execution time limit of 4 hours, and a 5 min grace period. We deployed full-sized `DeepSeek-R1` model with `sglang`, and accessed via the `litellm` API with generation parameters `temperature=0.6` and `top_p=0.95`.

For experiments done with MCTS, we set `num_children=5,` and `uct_c=0.25` unless stated otherwise. For evolutionary search, we set the number of `candidates_per_generation=5` for consistency. For AIRAMCTS, AIRAEVO, and AIRAGREEDY, we limit a debug cycle to total of 10 nodes or 12 hours of total time spent debugging (whichever comes first), to prevent spending all the resources on a single node.

## E.1   MCTS Statistics and Backup

Each node $v$ stores two running statistics: the visit count $N(v) \in \mathbb{N}$ and the empirical mean fitness $Q(v) \in \mathbb{R}$. When a freshly expanded leaf $v_\ell$ is evaluated, we initialize $N(v_\ell) = 1$ and $Q(v_\ell) = \mathcal{F}(v_\ell)$, where $\mathcal{F}$ is the proxy fitness used throughout the paper. The value $\mathcal{F}(v_\ell)$ is then *back-propagated* along the path $P = (v_\ell, \ldots, v_0)$ to the root:

$$\forall u \in P: \quad N(u) \leftarrow N(u) + 1, \qquad Q(u) \leftarrow Q(u) + \frac{\mathcal{F}(v_\ell) - Q(u)}{N(u)}.$$

This incremental update maintains the invariant $Q(u) = \frac{1}{N(u)} \sum_{i=1}^{N(u)} \mathcal{F}(v_i)$, i.e. $Q(u)$ is always the mean fitness of *all* leaf evaluations that have propagated through $u$. If a node is re-visited later, the same update is applied, allowing $Q$ to converge to the true expected value under continued exploration. No simulated roll-outs are performed; the leaf value is taken directly from $\mathcal{F}$.

Table 3: Per-task performance, grouped by complexity.

| Competition Name | Bronze Medal Rate (%) | Silver Medal Rate (%) | Gold Medal Rate (%) | Year | MLE-bench-30 |
|---|---|---|---|---|---|
| **LOW COMPLEXITY** | | | | | |
| detecting-insults-in-social-commentary | 100.0 ± 0.0 | 100.0 ± 0.0 | 100.0 ± 0.0 | 2012 | |
| the-icml-2013-whale-challenge-right-whale-redux | 100.0 ± 0.0 | 100.0 ± 0.0 | 85.0 ± 8.2 | 2013 | |
| mlsp-2013-birds | 70.0 ± 10.5 | 45.0 ± 11.4 | 0.0 ± 0.0 | 2013 | ✓ |
| random-acts-of-pizza | 40.0 ± 11.2 | 15.0 ± 8.2 | 0.0 ± 0.0 | 2015 | |
| denoising-dirty-documents | 90.0 ± 6.9 | 90.0 ± 6.9 | 90.0 ± 6.9 | 2015 | |
| text-normalization-challenge-russian-language | 65.0 ± 10.9 | 15.0 ± 8.2 | 0.0 ± 0.0 | 2017 | |
| dogs-vs-cats-redux-kernels-edition | 90.0 ± 6.9 | 90.0 ± 6.9 | 90.0 ± 6.9 | 2017 | |
| spooky-author-identification | 85.0 ± 8.2 | 85.0 ± 8.2 | 0.0 ± 0.0 | 2017 | ✓ |
| text-normalization-challenge-english-language | 50.0 ± 11.5 | 35.0 ± 10.9 | 0.0 ± 0.0 | 2017 | |
| leaf-classification | 45.0 ± 11.4 | 25.0 ± 9.9 | 5.0 ± 5.0 | 2017 | |
| jigsaw-toxic-comment-classification-challenge | 30.0 ± 10.5 | 5.0 ± 5.0 | 0.0 ± 0.0 | 2018 | |
| new-york-city-taxi-fare-prediction | 0.0 ± 0.0 | 0.0 ± 0.0 | 0.0 ± 0.0 | 2018 | ✓ |
| nomad2018-predict-transparent-conductors | 75.0 ± 9.9 | 70.0 ± 10.5 | 50.0 ± 11.5 | 2018 | ✓ |
| dog-breed-identification | 0.0 ± 0.0 | 0.0 ± 0.0 | 0.0 ± 0.0 | 2018 | |
| aerial-cactus-identification | 95.0 ± 5.0 | 95.0 ± 5.0 | 95.0 ± 5.0 | 2019 | |
| aptos2019-blindness-detection | 50.0 ± 11.5 | 40.0 ± 11.2 | 15.0 ± 8.2 | 2019 | ✓ |
| histopathologic-cancer-detection | 95.0 ± 5.0 | 95.0 ± 5.0 | 95.0 ± 5.0 | 2019 | |
| siim-isic-melanoma-classification | 0.0 ± 0.0 | 0.0 ± 0.0 | 0.0 ± 0.0 | 2020 | |
| plant-pathology-2020-fgvc7 | 95.0 ± 5.0 | 95.0 ± 5.0 | 95.0 ± 5.0 | 2020 | |
| ranzcr-clip-catheter-line-classification | 0.0 ± 0.0 | 0.0 ± 0.0 | 0.0 ± 0.0 | 2021 | |
| tabular-playground-series-dec-2021 | 35.0 ± 10.9 | 35.0 ± 10.9 | 35.0 ± 10.9 | 2021 | |
| tabular-playground-series-may-2022 | 0.0 ± 0.0 | 0.0 ± 0.0 | 0.0 ± 0.0 | 2022 | |
| **MEDIUM COMPLEXITY** | | | | | |
| facebook-recruiting-iii-keyword-extraction | 0.0 ± 0.0 | 0.0 ± 0.0 | 0.0 ± 0.0 | 2013 | |
| multi-modal-gesture-recognition | 0.0 ± 0.0 | 0.0 ± 0.0 | 0.0 ± 0.0 | 2013 | ✓ |
| billion-word-imputation | 5.0 ± 5.0 | 5.0 ± 5.0 | 5.0 ± 5.0 | 2015 | ✓ |
| cdiscount-image-classification-challenge | 0.0 ± 0.0 | 0.0 ± 0.0 | 0.0 ± 0.0 | 2017 | |
| statoil-iceberg-classifier-challenge | 0.0 ± 0.0 | 0.0 ± 0.0 | 0.0 ± 0.0 | 2018 | |
| tensorflow-speech-recognition-challenge | 0.0 ± 0.0 | 0.0 ± 0.0 | 0.0 ± 0.0 | 2018 | |
| tgs-salt-identification-challenge | 0.0 ± 0.0 | 0.0 ± 0.0 | 0.0 ± 0.0 | 2018 | |
| whale-categorization-playground | 35.0 ± 10.9 | 25.0 ± 9.9 | 0.0 ± 0.0 | 2018 | ✓ |
| champs-scalar-coupling | 0.0 ± 0.0 | 0.0 ± 0.0 | 0.0 ± 0.0 | 2019 | ✓ |
| inaturalist-2019-fgvc6 | 55.0 ± 11.4 | 45.0 ± 11.4 | 5.0 ± 5.0 | 2019 | |
| jigsaw-unintended-bias-in-toxicity-classification | 0.0 ± 0.0 | 0.0 ± 0.0 | 0.0 ± 0.0 | 2019 | ✓ |
| kuzushiji-recognition | 15.0 ± 8.2 | 0.0 ± 0.0 | 0.0 ± 0.0 | 2019 | ✓ |
| freesound-audio-tagging-2019 | 30.0 ± 10.5 | 10.0 ± 6.9 | 0.0 ± 0.0 | 2019 | ✓ |
| tensorflow2-question-answering | 0.0 ± 0.0 | 0.0 ± 0.0 | 0.0 ± 0.0 | 2020 | ✓ |
| imet-2020-fgvc7 | 0.0 ± 0.0 | 0.0 ± 0.0 | 0.0 ± 0.0 | 2020 | ✓ |
| google-quest-challenge | 95.0 ± 5.0 | 95.0 ± 5.0 | 45.0 ± 11.4 | 2020 | |
| herbarium-2020-fgvc7 | 70.0 ± 10.5 | 30.0 ± 10.5 | 0.0 ± 0.0 | 2020 | |
| osic-pulmonary-fibrosis-progression | 0.0 ± 0.0 | 0.0 ± 0.0 | 0.0 ± 0.0 | 2020 | ✓ |
| tweet-sentiment-extraction | 35.0 ± 10.9 | 35.0 ± 10.9 | 0.0 ± 0.0 | 2020 | ✓ |
| iwildcam-2020-fgvc7 | 60.0 ± 11.2 | 45.0 ± 11.4 | 20.0 ± 9.2 | 2020 | |
| alaska2-image-steganalysis | 0.0 ± 0.0 | 0.0 ± 0.0 | 0.0 ± 0.0 | 2020 | |
| plant-pathology-2021-fgvc8 | 100.0 ± 0.0 | 100.0 ± 0.0 | 95.0 ± 5.0 | 2021 | ✓ |
| cassava-leaf-disease-classification | 15.0 ± 8.2 | 10.0 ± 6.9 | 0.0 ± 0.0 | 2021 | ✓ |
| herbarium-2021-fgvc8 | 30.0 ± 10.5 | 0.0 ± 0.0 | 0.0 ± 0.0 | 2021 | |
| chaii-hindi-and-tamil-question-answering | 5.0 ± 5.0 | 5.0 ± 5.0 | 0.0 ± 0.0 | 2021 | |
| seti-breakthrough-listen | 65.0 ± 10.9 | 55.0 ± 11.4 | 45.0 ± 11.4 | 2021 | |
| hubmap-kidney-segmentation | 5.0 ± 5.0 | 5.0 ± 5.0 | 5.0 ± 5.0 | 2021 | ✓ |
| hotel-id-2021-fgvc8 | 85.0 ± 8.2 | 40.0 ± 11.2 | 0.0 ± 0.0 | 2021 | ✓ |
| ventilator-pressure-prediction | 0.0 ± 0.0 | 0.0 ± 0.0 | 0.0 ± 0.0 | 2021 | ✓ |
| AI4Code | 0.0 ± 0.0 | 0.0 ± 0.0 | 0.0 ± 0.0 | 2022 | |
| us-patent-phrase-to-phrase-matching | 55.0 ± 11.4 | 45.0 ± 11.4 | 15.0 ± 8.2 | 2022 | ✓ |
| uw-madison-gi-tract-image-segmentation | 0.0 ± 0.0 | 0.0 ± 0.0 | 0.0 ± 0.0 | 2022 | ✓ |
| h-and-m-personalized-fashion-recommendations | 10.0 ± 6.9 | 5.0 ± 5.0 | 0.0 ± 0.0 | 2022 | ✓ |
| herbarium-2022-fgvc9 | 5.0 ± 5.0 | 0.0 ± 0.0 | 0.0 ± 0.0 | 2022 | |
| petfinder-pawpularity-score | 35.0 ± 10.9 | 30.0 ± 10.5 | 15.0 ± 8.2 | 2022 | ✓ |
| icecube-neutrinos-in-deep-ice | 0.0 ± 0.0 | 0.0 ± 0.0 | 0.0 ± 0.0 | 2023 | |
| learning-agency-lab-automated-essay-scoring-2 | 25.0 ± 9.9 | 25.0 ± 9.9 | 25.0 ± 9.9 | 2024 | |
| lmsys-chatbot-arena | 0.0 ± 0.0 | 0.0 ± 0.0 | 0.0 ± 0.0 | 2024 | |
| **HIGH COMPLEXITY** | | | | | |
| iwildcam-2019-fgvc6 | 100.0 ± 0.0 | 100.0 ± 0.0 | 100.0 ± 0.0 | 2019 | |
| 3d-object-detection-for-autonomous-vehicles | 25.0 ± 9.9 | 15.0 ± 8.2 | 0.0 ± 0.0 | 2019 | |
| stanford-covid-vaccine | 65.0 ± 10.9 | 65.0 ± 10.9 | 65.0 ± 10.9 | 2020 | ✓ |
| bms-molecular-translation | 0.0 ± 0.0 | 0.0 ± 0.0 | 0.0 ± 0.0 | 2021 | ✓ |
| vinbigdata-chest-xray-abnormalities-detection | 0.0 ± 0.0 | 0.0 ± 0.0 | 0.0 ± 0.0 | 2021 | |
| predict-volcanic-eruptions-ingv-oe | 100.0 ± 0.0 | 100.0 ± 0.0 | 100.0 ± 0.0 | 2021 | |
| siim-covid19-detection | 0.0 ± 0.0 | 0.0 ± 0.0 | 0.0 ± 0.0 | 2021 | |
| rsna-miccai-brain-tumor-radiogenomic-classification | 30.0 ± 10.5 | 20.0 ± 9.2 | 5.0 ± 5.0 | 2021 | |
| rsna-2022-cervical-spine-fracture-detection | 0.0 ± 0.0 | 0.0 ± 0.0 | 0.0 ± 0.0 | 2022 | |
| smartphone-decimeter-2022 | 0.0 ± 0.0 | 0.0 ± 0.0 | 0.0 ± 0.0 | 2022 | ✓ |
| rsna-breast-cancer-detection | 0.0 ± 0.0 | 0.0 ± 0.0 | 0.0 ± 0.0 | 2023 | |
| google-research-identify-contrails-reduce-global-warming | 0.0 ± 0.0 | 0.0 ± 0.0 | 0.0 ± 0.0 | 2023 | |
| nfl-player-contact-detection | 5.0 ± 5.0 | 0.0 ± 0.0 | 0.0 ± 0.0 | 2023 | ✓ |
| vesuvius-challenge-ink-detection | 0.0 ± 0.0 | 0.0 ± 0.0 | 0.0 ± 0.0 | 2023 | |
| hms-harmful-brain-activity-classification | 0.0 ± 0.0 | 0.0 ± 0.0 | 0.0 ± 0.0 | 2024 | ✓ |

## E.2 Draft

You are a Kaggle Grandmaster attending a high-stakes competition.
Carefully consider the task description, the size and format of the available data,
↪   as well as the available compute resources.
Your goal is to provide EXACTLY ONE IDEA AND ONE CODE IMPLEMENTATION of the idea,
↪   different from those previously explored, that leverages the available resources
↪   and is likely to lead to strong performance on the competition.
Be specific about each step of the proposed approach, including data processing and
↪   feature engineering, the modeling and optimization method, as well as the
↪   evaluation (USE 5-FOLD CROSS-VALIDATION).
You MUST PROVIDE a solution IDEA/PLAN in natural language and CODE in python that
↪   DOES NOT INVOLVE any exploratory data analysis.
# TASK DESCRIPTION
````
{{task_desc}}
````

# PREVIOUSLY EXPLORED IDEAS
````markdown
{{memory}}
````

# DATA OVERVIEW
````
{{data_overview}}
````

**CONSTRAINTS**:
  - Be aware of the running time of the solution, it should complete within
  ↪   {{execution_timeout}}
  - Prefer vectorized operations over Python loops when processing large datasets.
  - Use `torch.optim.AdamW` (the recommended optimizer) instead of the deprecated
  ↪   `AdamW` from `transformers`.
  - Replace the deprecated `early_stopping_rounds` argument in `lightgbm.train()`
  ↪   with the `lightgbm.early_stopping(stopping_rounds=...)` callback.
  - If using `timm` models, remember not to prefix or suffix the model names with
  ↪   datasets such as `cifar` as this was deprecated.
  - As much as possible, keep the stdout clean.
**DATA**: The data is already prepared and available in the read-only `./data`
↪   directory. You should not unzip any files.
**COMPUTE**: You have access to a Python environemnt with 1 NVIDIA H200 GPU(s) and
↪   24 CPUs available, and the following packages installed: {{packages}}. If you
↪   need to, feel free to use additional libraries that fit the problem.
Consider the previously explored ideas, and make sure the idea you propose considers
↪   a DIFFERENT ASPECT OF THE SOLUTION, but keep the EVALUATION CONSISTENT.
Brainstorm about possible approaches and WHY THEY ARE LIKELY TO BE EFFECTIVE AND
↪   INCREASE THE PERFORMANCE for the given task, and the available data and compute
↪   resources.
Remember, and this is important, the first idea should be simple and easy to
↪   implement, while the last one should be more complex and sophisticated.
{% if draft_complexity == 'simple' %}
In this iteration **focus on PROPOSING A SIMPLE IDEA:** one that can serve as a
↪   SIMPLE YET EFFECTIVE BASELINE for the task. For example, consider battle-tested
↪   methods or (potentially pre-trained) models that are known to work well for the
↪   task at hand.
{% elif draft_complexity == 'normal' %}
In this iteration **focus on PROPOSING A MORE COMLPEX IDEA:** one that can beat the
↪   previous baselines at the cost of some complexity and compute. For example,
↪   consider leveraging more complex and/or larger (potentially pre-trained) models,
↪   specialized feature engineering, or basic ensambling and/or hyper-parameter
↪   optimization.
{% elif draft_complexity == 'complex' %}
In this iteration **focus on PROPOSING AN ADVANCED IDEA:** one that can beat the
↪   previous baselines at the cost of some complexity and compute. For example,
↪   consider using specialized (potentially pre-trained) models, leveraging advanced
↪   feature engineering or data augmentiation strategies, advanced ensambling and/or
↪   hyper-parameter optimization.

```
{% endif %}
**RESPONSE FORMAT FOR IMPLEMENTATION**:
Provide a **SINGLE** Markdown code block (wrapped in ```) for the implementation
↪  containing a **SELF-CONTAINED** Python script that:
1. Implements the idea **END-TO-END**
2. **PRINTS THE 5-FOLD CROSS-VALIDATION** score of the evaluation metric
3. **SAVES THE TEST PREDICTIONS** in a `submission.csv` file in the current
↪  directory
Start by making sure you understand the task, the data and compute resources and the
↪  idea. Then generate a detailed implementation plan that will structure and guide
↪  you step-by-step through the implementation process. Make sure to reflect on the
↪  plan to ensure that the implementation is efficient and faithful to the idea,
↪  and that all the requirements (e.g., the evaluation score is printed, the
↪  submission file follows the correct format and is saved in the correct location,
↪  etc.) are satisfied.
For large datasets, avoid for loops and aim for efficient and fast data loading and
↪  feature engineering.
Format the proposed solution as follows:
# Idea to implement
<the proposed idea/plan>
```python
<the implementation of the proposed idea/plan>
```
```

## E.3   Improve

```
# Introduction:
You are a Kaggle Grandmaster attending a high-stakes competition.
Carefully consider the task description, the size and format of the available data,
↪  as well as the available compute resources.
Your goal is to provide EXACTLY ONE IDEA AND ONE CODE IMPLEMENTATION of the idea,
↪  different from those previously explored, that improves upon an existing
↪  solution to the task.
Be specific about each step of the proposed improvement, including data processing
↪  and feature engineering, the modeling and optimization method, as well as the
↪  evaluation (USE 5-FOLD CROSS-VALIDATION).
You MUST PROVIDE an improvement IDEA/PLAN in natural language and CODE in python
↪  that DOES NOT INVOLVE any exploratory data analysis.
# TASK DESCRIPTION
````
{{task_desc}}
````
# PREVIOUS SOLUTION:
## Code:
{{prev_code}}
## Terminal Output:
{{prev_terminal_output}}
# PREVIOUSLY EXPLORED IMPROVEMENT IDEAS
````markdown
{{memory}}
````
# DATA OVERVIEW
````
{{data_overview}}
````
**CONSTRAINTS**:
  - Be aware of the running time of the solution, it should complete within
  ↪  {{execution_timeout}}
  - Prefer vectorized operations over Python loops when processing large datasets.
  - Use `torch.optim.AdamW` (the recommended optimizer) instead of the deprecated
  ↪  `AdamW` from `transformers`.
```

- Replace the deprecated `early_stopping_rounds` argument in `lightgbm.train()`
  ↪ with the `lightgbm.early_stopping(stopping_rounds=...)` callback.
- If using `timm` models, remember not to prefix or suffix the model names with
  ↪ datasets such as `cifar` as this was deprecated.
- As much as possible, keep the stdout clean.
**DATA**: The data is already prepared and available in the read-only `./data`
↪ directory. You should not unzip any files.
**COMPUTE**: You have access to a Python environemnt with 1 NVIDIA H200 GPU(s) and
↪ 24 CPUs available, and the following packages installed: {{packages}}. If you
↪ need to, feel free to use additional libraries that fit the problem.
Consider the previously explored ideas, and make sure the improvement idea you
↪ propose considers a DIFFERENT IMPROVEMENT OF THE SOLUTION, but keep the
↪ EVALUATION CONSISTENT.
Brainstorm about possible improvements and WHY THEY ARE LIKELY TO BE EFFECTIVE AND
↪ INCREASE THE PERFORMANCE for the given task, and the available data and compute
↪ resources.
{% if improve_complexity == 'simple' %}
In this iteration, suggest a *minimal, low-risk* tweak that keeps the current
↪ solution's core intact-no architecture overhauls or fundamental methodology
↪ changes.
Think: a feature-engineering twist, a lightweight data-augmentation trick, or
↪ hyperparameter changes.
Check the MEMORY section first and avoid duplicating earlier ideas.
{% elif improve_complexity == 'normal' %}
In this iteration, propose a *moderate upgrade* that builds on the baseline without
↪ deviating dramatically.
Options include (but not limited to) hyper-parameter tuning, a small ensemble of
↪ similar models, a sturdier preprocessing pipeline, feature engineering
↪ improvements, and data augmentation.
Check the MEMORY section first and avoid duplicating earlier ideas.
{% elif improve_complexity == 'complex' %}
In this iteration, recommend a *substantial extension* that pushes the method's
↪ boundaries while preserving its core logic.
Consider advanced ensembling/stacking, fine-tuning specialized pre-trained models,
↪ or exhaustive hyper-parameter searches.
Check the MEMORY section first and avoid duplicating earlier ideas.
{% endif %}
**RESPONSE FORMAT FOR IMPLEMENTATION**:
Provide a **SINGLE** Markdown code block (wrapped in ```) containing a
↪ **SELF-CONTAINED** Python script that:
1. Implements the idea **END-TO-END**
2. **PRINTS THE 5-FOLD CROSS-VALIDATION** score of the evaluation metric
3. **SAVES THE TEST PREDICTIONS** in a `submission.csv` file in the current
↪ directory
Start by making sure you understand the task, the data and compute resources and the
↪ idea. Then generate a detailed implementation plan that will structure and guide
↪ you step-by-step through the implementation process. Make sure to reflect on the
↪ plan to ensure that the implementation is efficient and faithful to the idea,
↪ and that all the requirements (e.g., the evaluation score is printed, the
↪ submission file follows the correct format and is saved in the correct location,
↪ etc.) are satisfied.
For large datasets, avoid for loops and aim for efficient and fast data loading and
↪ feature engineering.
Format the proposed solution as follows:
# Improvement Idea to implement
<the proposed improvement idea/plan>
```python
<the implementation of the proposed improvement>
```

## E.4 Analysis

```
# Introduction:

You are a Kaggle grandmaster attending a competition.

You have written code to solve this task and now need to evaluate the output
of the code execution.

You should determine if there were any bugs as well as report the empirical
findings.

# Task Description:

{{task_desc}}

# Implementation:

{{code}}

# Execution output:

{{execution_output}}
```

## E.5 Debug

```
# Introduction:
You are a Kaggle Grandmaster fixing code bugs in a high-stakes competition
↪   solution.
Carefully review the previous debugging attempts, the buggy code and its
↪   terminal output in addition to the given task/data details, and
↪   available compute resources.
You must not change the core idea or methodology of the solution, but only
↪   fix the bugs in the code.
# Task Description:
````markdown
{{task_desc}}
````
{% if memory %}
# Previous debugging attempts:
````markdown
{{memory}}
````
{% endif %}
# Buggy Implementation:
{{prev_buggy_code}}
# Execution Output (Error):
{{execution_output}}
# Data Overview:
````
{{data_overview}}
````
# Compute Environment:
```

```
- GPU: 1 NVIDIA H200
- CPUs: 24
- Available Packages: {{packages}}
- Additional libraries allowed as needed.
# Instructions:
- **Do NOT** alter the core method or underlying idea. Only correct
↪  existing bugs.
- Outline your bug-fix plan clearly in 3-5 concise sentences.
- Provide a single, complete Python code block wrapped in markdown
↪  (```python) that:
- Implements the fix fully.
- Calculates and clearly prints the evaluation metric using a validation
↪  set (use 5-FOLD CV if suitable).
- Generates a `submission.csv` file with test set predictions stored in the
↪  **current directory** (`./submission.csv`).
- Is fully self-contained and executable as-is (The entire bug-free
↪  solution is given).
- **Important Reminders:**
- Absolutely do **NOT** skip any part of the code.
- Always ensure predictions on the provided unlabeled test set are saved in
↪  `./submission.csv`. This is crucial for grading.
# Other remarks
- Huggingface is set to OFFLINE mode by default. If you firmly believe that
↪  the issue is not having the requested model in the cache, please set it
↪  to ONLINE mode by setting both the environment variables
↪  `HF_HUB_OFFLINE=0` and `TRANSFORMERS_OFFLINE=0` on top of your code, by
↪  importing and using `os.environ[...] = ...`.
- Do not set/force Huggingface to OFFLINE mode as that will NOT fix any
↪  issue.
- When a model cannot be found in the `timm` library, it might be useful to
↪  `print(timm.list_models())`.
- If using `timm` models, remember not to prefix or suffix the model names
↪  with datasets such as `cifar` as this was deprecated.
Brainstorm about possible ways to fix the bug and WHY THEY ARE LIKELY TO
↪  FIX THE BUG for the given implementation. Additionally, if any other
↪  bugs further down the line are observed, please fix them as well.
Generate a bug-fix plan that will structure and guide your step-by-step
↪  reasoning process. Reflect on it to make sure all the requirements are
↪  satisfied.
Format the proposed bug-fix plan and code as follows:
# Bug Fix Plan
<bug-fix plan>
```python
<the fixed python code>
```
```

## E.6  Crossover

**CROSSOVER Operator**  For two or more valid artifacts, the Crossover operator performs a merge of existing solutions. It specifically aims to create a solution drawing insights from the best aspects of its parents, while also taking into account the shared task context (see Section 2.3) that informs and guides the merging process. For each crossover operation, the two parent nodes are sampled from the distribution of all nodes' fitness scores.

```
# Introduction:
```

```
You are a Kaggle Grandmaster attending a high-stakes competition.
Your goal is to combine two previously developed solutions to further
↪  increase performance on the given task.
Carefully consider the task description, the two provided solutions,
↪  the available data, and compute resources.
You need to devise a plan to merge or integrate these solutions and
↪  then implement it.

# TASK DESCRIPTION
```
{{task_desc}}
```

{% if memory %}
# PREVIOUSLY EXPLORED CROSSOVER/COMBINATION IDEAS
```markdown
{{memory}}
```
{% endif %}

# PREVIOUS SOLUTION 1:
## Code:
```python
{{prev_code1}}
```

# PREVIOUS SOLUTION 2:
## Code:
```python
{{prev_code2}}
```

# INSTRUCTIONS:

Your main task is to:
1. Propose a **Crossover Plan** in natural language explaining how to
↪  combine Solution 1 and Solution 2.
2. Provide a **Python Code Implementation** of this plan.

Consider any previously explored ideas from the MEMORY section.
Brainstorm how the two provided solutions can be effectively combined
↪  and **WHY THIS CROSSOVER IS LIKELY TO BE EFFECTIVE** and increase
↪  performance for the given task, data, and compute resources.
Aim for a consistent evaluation method (e.g., 5-FOLD CROSS-VALIDATION,
↪  unless the task specifics dictate otherwise).

**CONSTRAINTS**:
- Be aware of the running time of the solution, it should complete
↪  within {{execution_timeout}}
- Prefer vectorized operations over Python loops when processing large
↪  datasets.
- Use `torch.optim.AdamW` (the recommended optimizer) instead of the
↪  deprecated `AdamW` from `transformers`.
- Replace the deprecated `early_stopping_rounds` argument in
↪  `lightgbm.train()` with the
↪  `lightgbm.early_stopping(stopping_rounds=...)` callback.
- If using `timm` models, remember not to prefix or suffix the model
↪  names with datasets such as `cifar` as this was deprecated.
```

```
        - As much as possible, keep the stdout clean.

    **DATA**: The data is already prepared and available in the read-only
    ↪  `./data` directory. You should not unzip any files.

    **COMPUTE**: You have access to a Python environment with 1 NVIDIA H200
    ↪  GPU(s) and 24 CPUs available, and the following packages installed:
    ↪  {{packages}}. If you need to, feel free to use additional libraries
    ↪  that fit the problem.

    Start by making sure you understand the task, the provided solutions,
    ↪  the data and compute resources, and your proposed crossover idea.
    ↪  Then, generate a detailed internal implementation plan that will
    ↪  structure and guide you step-by-step through the implementation
    ↪  process. Make sure to reflect on the plan to ensure that the
    ↪  implementation is efficient, faithful to the crossover idea, and
    ↪  that all requirements (e.g., the evaluation score is printed, the
    ↪  submission file follows the correct format and is saved in the
    ↪  correct location, etc.) are satisfied.

    **RESPONSE FORMAT FOR IMPLEMENTATION**:
    Provide a **SINGLE** Markdown code block (wrapped in ```) containing a
    ↪  **SELF-CONTAINED** Python script that:
    1. Implements the idea **END-TO-END**
    2. **PRINTS THE 5-FOLD CROSS-VALIDATION** score of the evaluation
    ↪  metric
    3. **SAVES THE TEST PREDICTIONS** in a `submission.csv` file in the
    ↪  current directory

    Format the proposed solution as follows:

    # Crossover Plan
    <Your proposed crossover plan/strategy>

    ```python
    <the implementation of the crossover solution>
    ```
```

## F  Completion Tokens Per Method

In Fig. 10, we summarize each method's average number of completion tokens per operator. The analysis suggests that our changes to the operators (see Section 4.1) lead to substantially longer thinking chains.

## G  Infrastructure Lessons

The infrastructure design of AIRA-dojo was informed by a few reliability and performance constraints:

1. **LLM Service.** As we scaled experiments, we observed that external LLM services slow down, risking timeouts. While the exact rate limits are set by the API provider [3], their existence nevertheless places an upper bound on the number of parallel actors and introduces a point of failure. Self-hosting LLMs is therefore required for reliable scaling.

---

[3] https://platform.openai.com/docs/guides/rate-limits

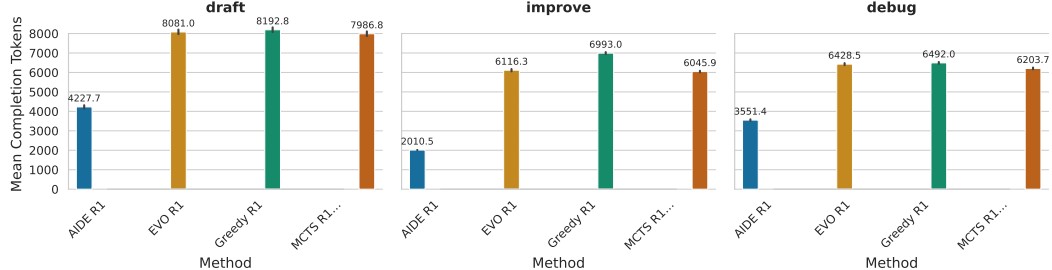

Figure 10: **Number of completion tokens per operator for each agent.** Each value is averaged across the independent runs on the MLE-bench lite suite in the main experiments (see Section 5.2).

2. **Environments.** Early experiments resulted in agents corrupting Python environments e.g., via a `pip install`. Virtualization technology, such as containers, was therefore a natural fit for state isolation.

3. **Checkpointing.** In line with prior work [22, 10, 25, 9], we observe that both hard (machine failure, filesystem failures) and soft failures (slowdowns) are commonplace. Consider, for example, 10 experiments, each with 100 agents and each running for 24 hours—that results in $10 \times 100 \times 24 = 24000$ hours of required uninterrupted uptime, which is significant when compared to a Mean Time to Failures of $\sim 1000$ hours per node [25]. We therefore introduced checkpointing support to mitigate the effects.

## H  Rationale for Selection of Search Policies

To systematically evaluate search policies we selected three complementary approaches. AIDE's greedy tree-search policy serves as an efficient baseline that prioritizes exploitation. Monte Carlo Tree Search (MCTS) extends this by allowing us to directly modulate the exploration-exploitation tradeoff through a single parameter. In contrast, the evolutionary graph-based search policy leverages population-based sampling and recombination, representing a fundamentally different strategy from both greedy and tree-based methods.

## I  Limitations and Future Work

There are several important dimensions for performance that we leave for future work to explore.

**Agentic operators.** Our experiments suggest that the effectiveness of search is heavily dependent on the capability of the operators. To manage complexity, in this work we experiment with LLM-based operators. However, one could readily use full-fledged agents as operators. For example, a natural extension would be to include an ideation agent as an operator [40], and to replace the implementation and debugging operators with a SWE-Agent [30, 50].

**Finetuning.** LLMs are critical components of the operators. Future work could investigate supervised fine-tuning or reinforcement learning as methods to enhance operator effectiveness.

**Scaling the search.** To maintain comparability with MLE-bench, we adopt the benchmark's time (24-hour limit) and compute constraints (1 GPU). However, solving challenging problems likely requires substantially greater resources, and evaluating performance under these restrictions provides limited insights (see Section C for an experiment extended to 120 hours—5 days). Studying the scaling behavior of search policies and operators, and developing agents capable of effectively leveraging more computational resources over longer time horizons, is an important direction for future research.

**Data contamination.** Finally, there is the issue of data contamination. It is possible that our results are influenced by the presence of information related to the evaluated Kaggle tasks, or similar tasks, within the LLM training data. Creating a continuous stream of fresh and novel tasks remains an important research challenge [47].

## J   Variance in the Performance Estimation

AI agent performance on complex benchmarks like MLE-bench exhibits substantial variance that can severely impact the reliability of comparative evaluations. While MLE-bench's official recommendations suggest using at least 3 seeds for evaluation, this may be insufficient for reliable agent rankings and performance estimation, particularly given that agents/LLMs can be quite high-variance inherently.

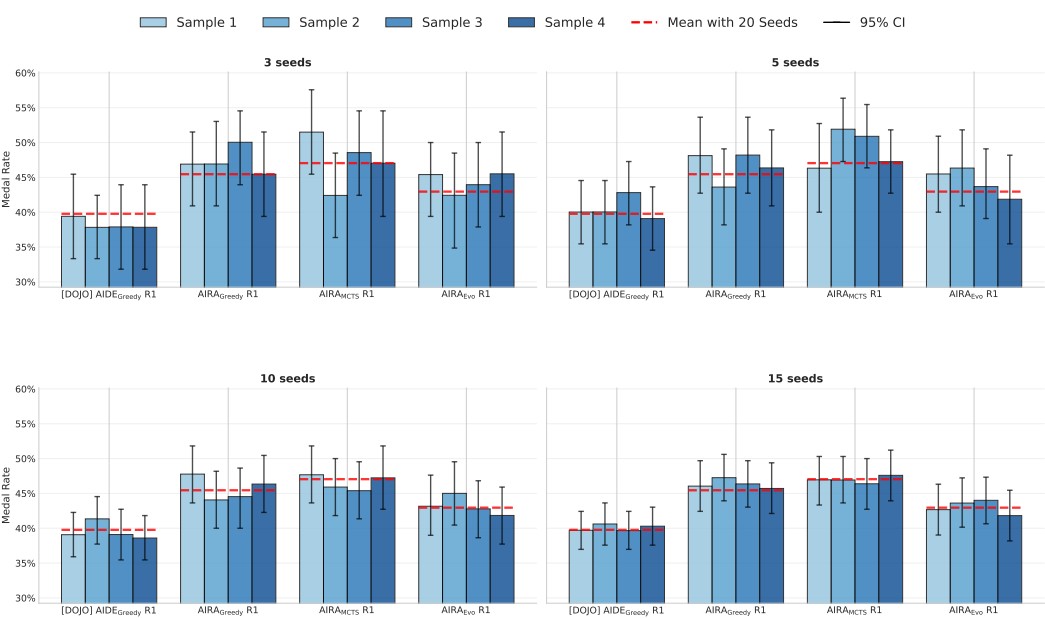

Figure 11: **Potential algorithm rankings under different seed counts.** This figure demonstrates how the same algorithms could be ranked differently if fewer seeds were used, illustrating the instability that results from estimating performance with insufficient samples. While confidence intervals theoretically capture this uncertainty, researchers often underestimate how dramatically rankings can shift with limited seeds. Each panel shows plausible alternative rankings that could emerge from the same underlying algorithm performance distributions. We recommend using a minimum of 10 seeds per task for moderate ranking stability, with 20 seeds preferred to avoid misleading conclusions about relative algorithm performance.

Figure 11 demonstrates how dramatically agent rankings can shift with insufficient seed sampling. Each run represents a sample from the underlying performance distribution (estimated from 20 seeds), and with only a few seeds, observed performance differences may be statistical artifacts rather than genuine capability differences. Given MLE-bench's computational intensity—with 75 competitions requiring substantial resources per run—most researchers face a critical trade-off. If options were to evaluate on all 75 competitions with 3 seeds each or evaluate on 22 competitions with 10 seeds each, the latter provides more reliable conclusions. While the former experimental setup offers broader coverage, the individual competition results are unreliable, making it difficult to determine whether an agent genuinely excels at specific types of ML engineering tasks. Thus, we prefer to enable confident identification of an agent's strengths and weaknesses across a representative subset of competitions.

For future work we recommend a minimum of 10 seeds per competition (with 20 seeds much more preferred), stratified bootstrapping for confidence intervals rather than standard error estimates, and a focus on competition subsets (e.g., MLE-bench Lite) with higher seed counts rather than full evaluation with few seeds.

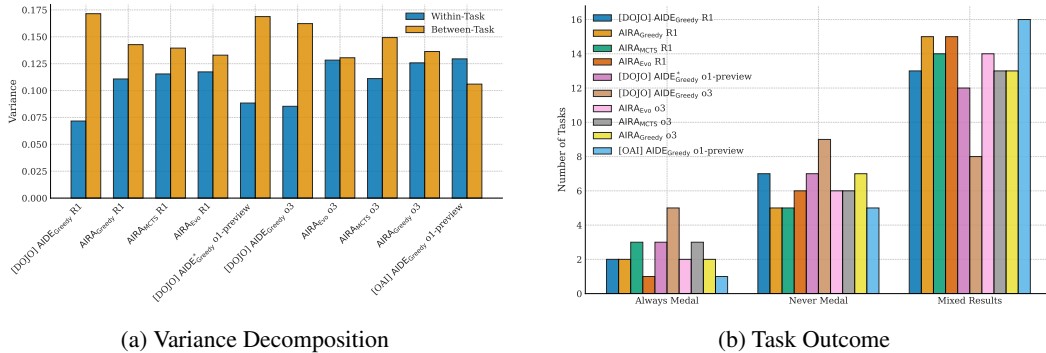

(a) Variance Decomposition                    (b) Task Outcome

Figure 12: **Sources of performance variance across tasks and seeds. (a)** Variance decomposition shows that a substantial portion of the observed variance in agent performance arises from between-task variability (across tasks). However, certain methods' medal achievements are equally as variable within a task. **(b)** Distribution of medal outcomes across tasks for each agent, showing how frequently a method consistently performs well (always medals), consistently fails (never medals), or exhibits inconsistent performance (sometimes medals). These figures underscore that agent evaluation on MLE-bench is impacted both by stochasticity in task-level performance and by systematic variation in task difficulty or agent specialization.

## K   Test-Validation Gap Results

In Fig. 4a, we present the performance profile of AIDEGREEDY. In Fig. 13, we show the performance profiles of AIRAMCTS, AIRAEVO, and AIRAGREEDY. Compared to Fig. 4a, the agents using AIRA operators display a smaller gap between test and validation scores.

## L   Per-Task Results

This section presents the non-aggregated, per-task results obtained from the MLE-bench Lite suite in Section 5.2. The results are summarized in Fig. 14.

## M   Sample Search Trees

We present samples of search trees from various methods used in the `spooky-author-identification` task, as shown in Figures 15 to 17. The colors of the nodes indicate the validation scores, while the labels within the nodes display the test scores. We represent medal-winning nodes with medal emojis and above-median ranking solutions (relative to the human leaderboard) with an ok emoji. The nodes in red are buggy nodes.

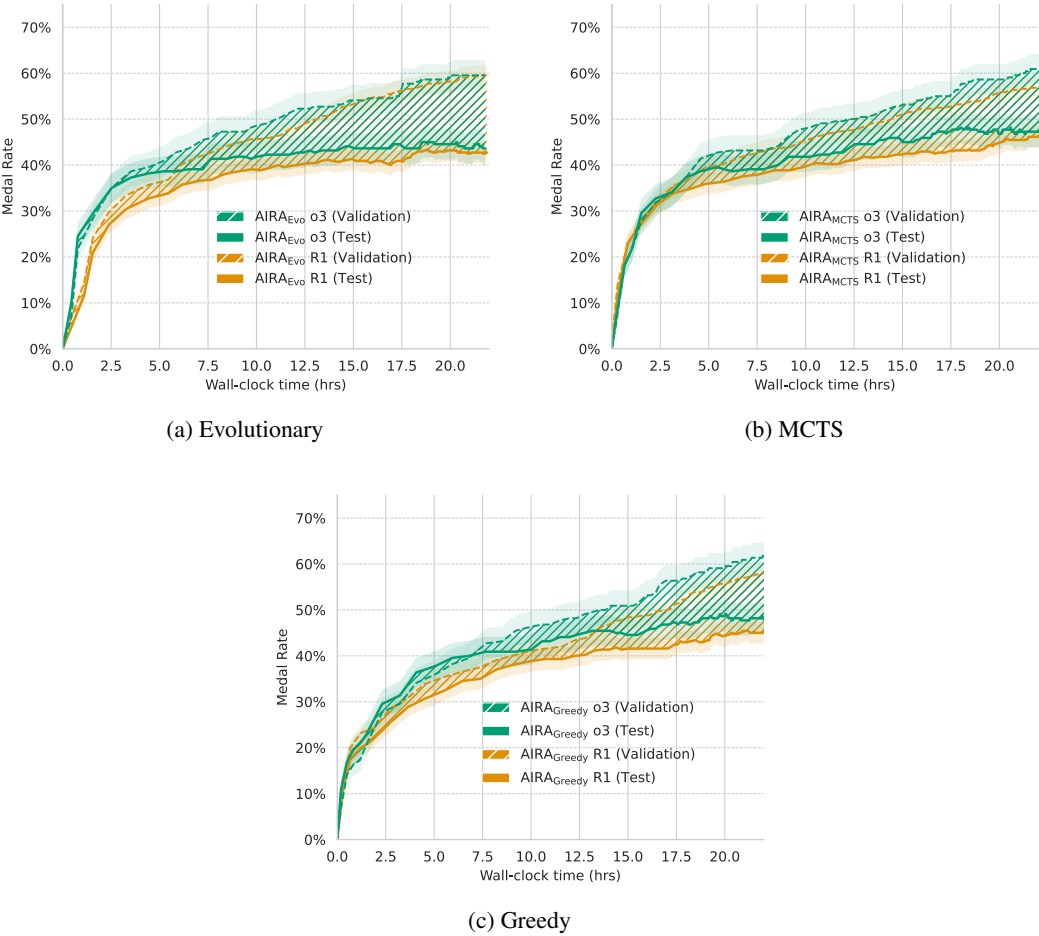

(a) Evolutionary

(b) MCTS

(c) Greedy

Figure 13: **Perceived vs. actual medal rate over 24 hours of searching with the AIRA operators using different search policies.** The curves show the mean validation (agent-reported) and held-out test medal rates across 20 seeds with R1 and 10 seeds o3 for all tasks. The widening band illustrates the generalization gap, revealing how apparent gains on the validation set can mask overfitting and ultimately undermine the search process.

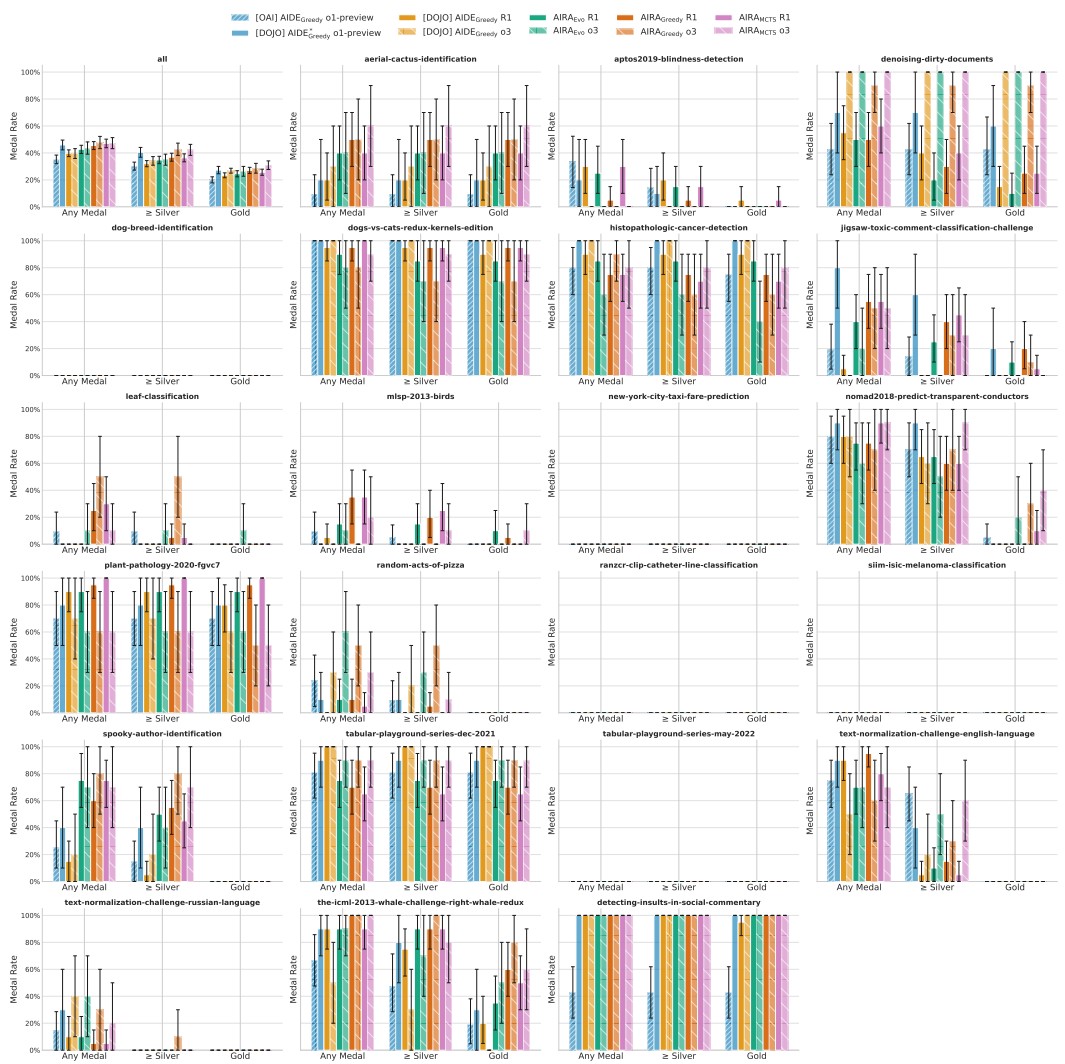

Figure 14: **Per-task performance.** Methods utilizing R1 are averaged over 20 seeds per task and methods utilising o3 are averaged over 10 seeds per task

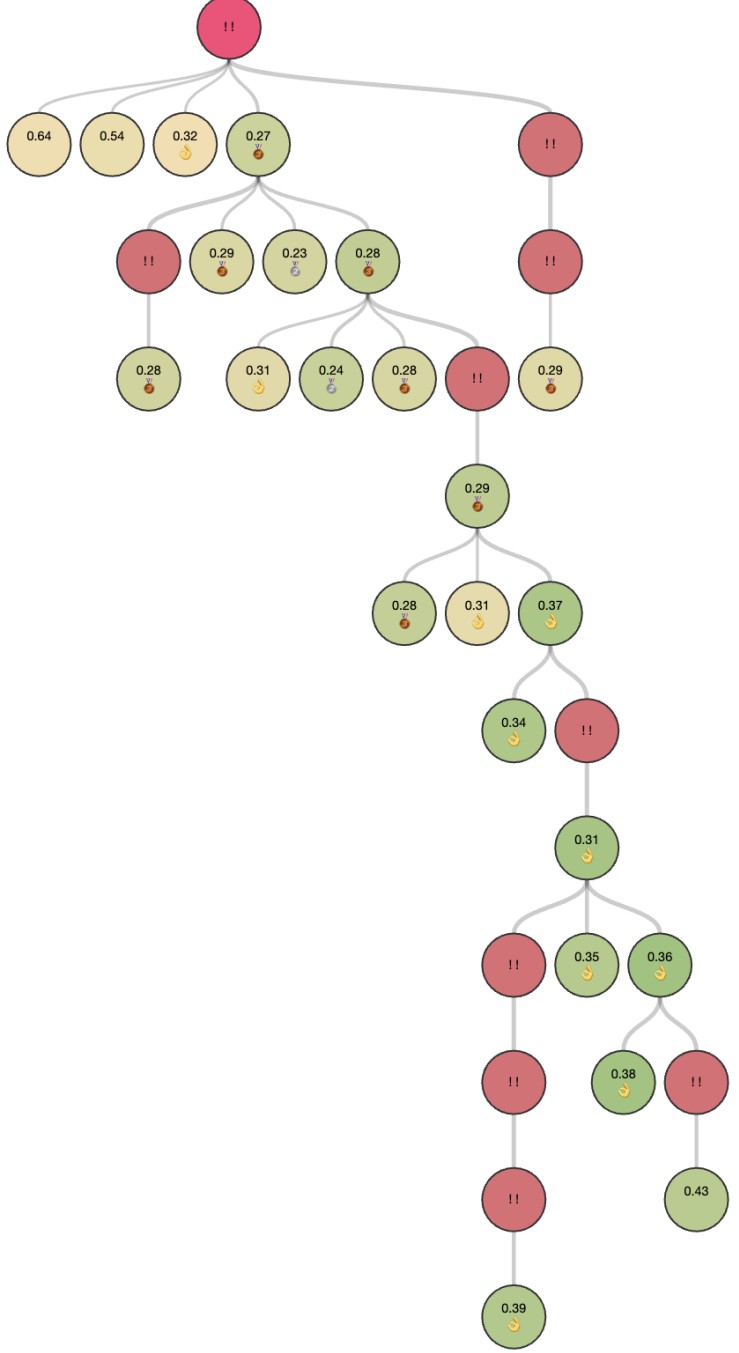

Figure 15: AIRAGREEDY

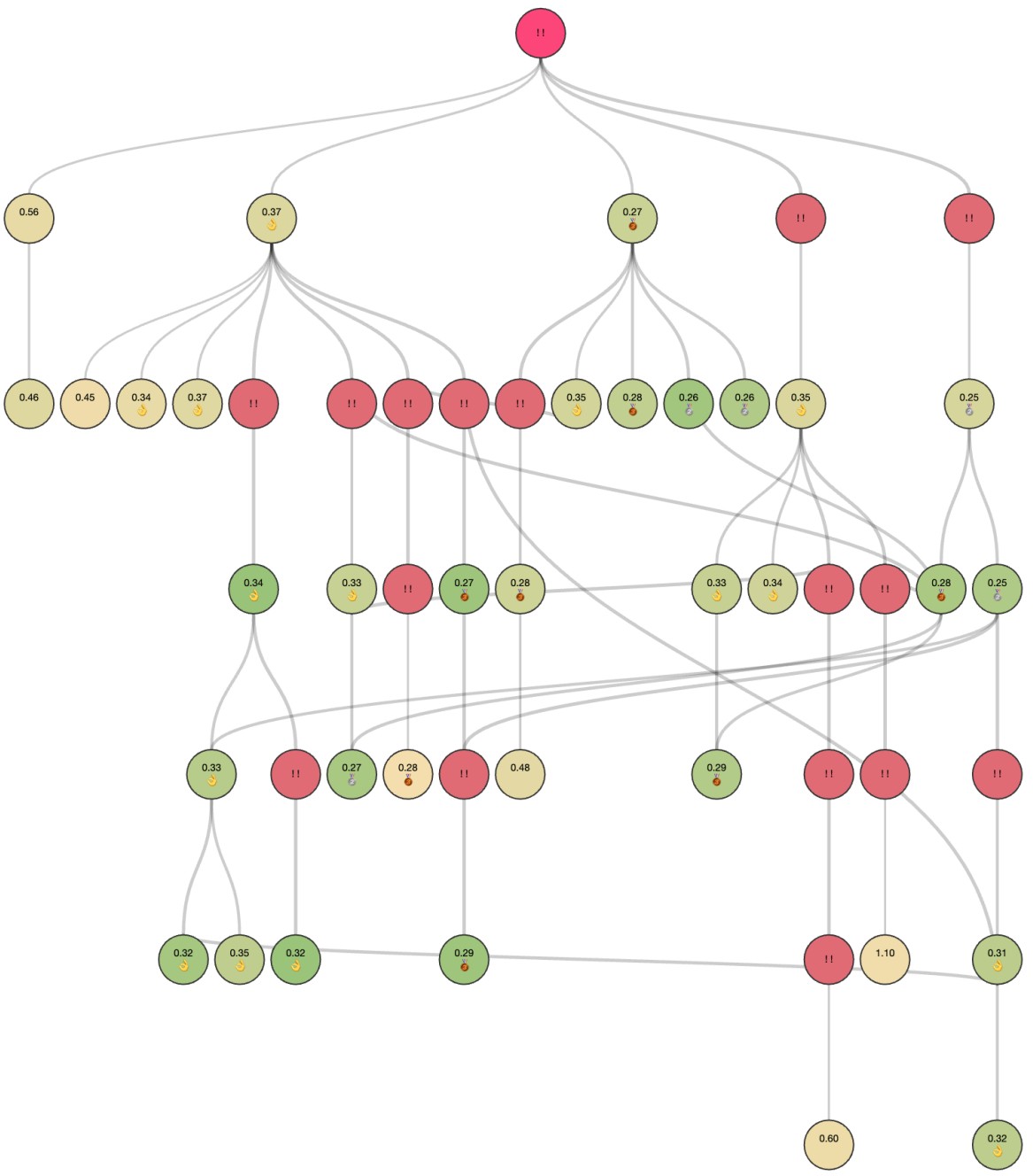

Figure 16: AIRAᴇᴠᴏ

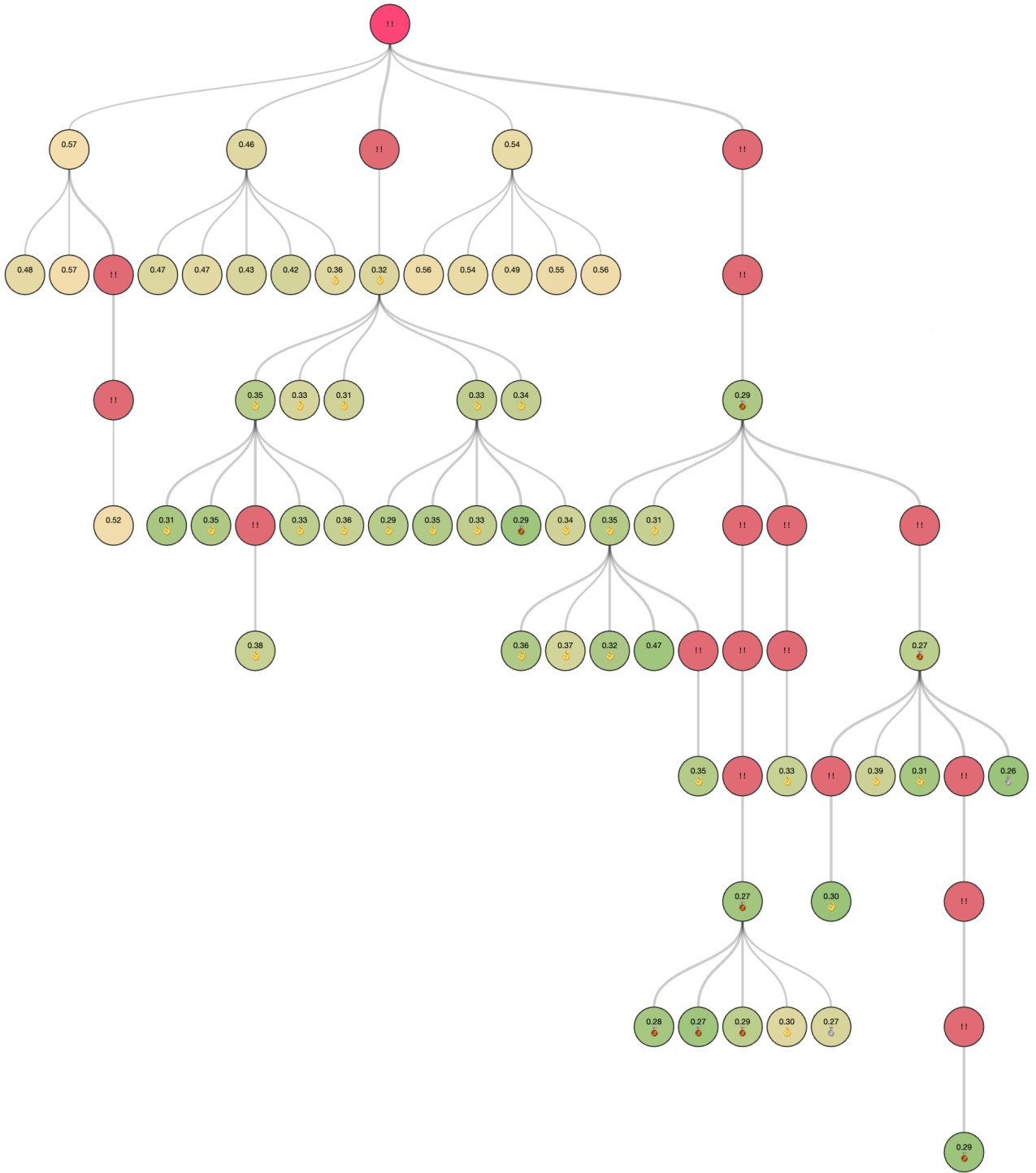

Figure 17: AIRA MCTS

