# OpenReview forum: "AI Research Agents for Machine Learning: Search, Exploration, and Generalization in MLE-bench"
_NeurIPS.cc/2025/Conference — NeurIPS 2025 spotlight_

### Official Review · Reviewer_DZn3 · 2025-07-01

**Clarity:** 3
**Significance:** 3
**Originality:** 2
**Rating:** 5
**Confidence:** 3

**Summary:**

The paper proposes a framework to construct LLM-based code agents for solving MLE-bench tasks. The primary contributions of the paper are algorithmic and empirical. Key contributions include the computational environment for agents (AIRA-Dojo), the agent design framework (AIRA) and the experimental results showing improved performance on MLE-bench (lite). AIRA builds on ideas in AIDE. The main technical ideas involve inducing a search space over programs using LLM-based operators like DRAFT, DEBUG, IMPROVE, MEMORY, etc. The sandbox infrastructure creates the environment to safely and reliably conduct the (long-running) search. Experiments demonstrate a new state of the art on the benchmark. The paper also includes a careful empirical study of the search algorithm's design choices.

**Questions:**

1. What is the performance of SOTA reasoning and code LLMs (that only do test-time search) on this task? How does it compare to AIDE and AIRA?

2. What kinds of mistakes or errors were observed in the programs generated on test datasets? What is the subjective and / or objective code quality?

3. What is the implementation of CROSSOVER from Line 106? I was unable to find it in the paper or appendix.

4. On Line 184, what figures are being referenced? ("In the figures, we take ..."). (I think it refers to the prior AIDE eval metrics reported in the graphs but it was confusing on first reading.)

**Ethical Concerns:**

["NO or VERY MINOR ethics concerns only"]

**Final Justification:**

I thank the authors for their detailed responses to the reviewers. The authors have addressed my main questions satisfactorily. After reading the other reviews and comments, I did not see any new significant technical issues raised in the other reviews. Since the authors have agreed to include the error analysis in the final version, I remain positive about the paper and continue to recommend acceptance.

**Limitations:**

Yes.

**Paper Formatting Concerns:**

None.

**Quality:**

3

**Strengths And Weaknesses:**

**Strengths**

- The paper studies an important and interesting problem. Progress here would have a large impact and be of interest to the community.
- The paper is reasonably well written. Key concepts are illustrated nicely.
- The proposed algorithmic approach is intuitively clear and technically sound. I wasn't able to find any major issues with the technical setup. The induced search space is clearly described and the operators well described in the appendix (with the exception of CROSSOVER).
- The experiments are well-constructed and nicely detailed. The primary result (Figure 1, Figure 3) shows that AIRA performs better than the previous SOTA (AIDE) on the MLE-bench lite benchmark. The careful investigation into the underlying components of performance in Section 5.2 and 5.3 contains valuable insights into the underlying performance factors. The appendix contains a good amount of additional detail.

**Weaknesses**

- The paper discusses its limitations reasonably well in Sec 7.1 I agree that leakage of the benchmark during the (unseen) base LLM training might be a concern. While AIDE is a good choice of baseline, I was unable to find any evaluations of SOTA reasoning and code LLMs (Gemini 2.5, Opus 4, etc.). (This would be zero-shot, without any search beyond the TTS used by the LLM itself during inference). While I don't expect to see outperformance, it would strengthen the paper's empirical standing and make it easier to assess its relative impact in the broader (and rapidly growing) area of using LLMs to construct search-based agents.

- The algorithmic novelty is a little low. The paper mostly builds on the ideas in AIDE and uses well-known planning algorithms. However, this is a minor issue given the careful and detailed empirical investigation.

- The paper does not include an error analysis of the generated programs or any code quality metrics. It would be useful to better understand the error distribution during program generation (beyond those having to do with inference infrastructure).

---

> ### Author Rebuttal · Authors · 2025-07-31
>
> Thank you for your thoughtful and constructive review. We appreciate your recognition that our work "studies an important and interesting problem" with "large impact" and that our approach is "technically sound."
>
> We're also grateful for your positive assessment that our "experiments are well-constructed and nicely detailed" and that our investigation into the underlying components "contains valuable insights."
>
> ### **Response to Major Questions**
>
> **Q1: Performance of SOTA reasoning and code LLMs as zero-shot baselines**
>
> This is an excellent point for contextualizing our agent's performance gains. We have now evaluated o3 for all search and operator combinations. We can extract a model’s single-shot performance by using the first node of each search tree/graph, which represents the initial DRAFT operation without any iterative refinement.
>
> | Method              | Medal Rate |
> |---------------------|------------|
> | o3 Single-shot      | 11.36%     |
> | R1 Single-shot      | 11.44%     |
> | [OAI]  AIDE_GREEDY  | 35.00%     |
> | [DOJO] AIRA_MCTS    | 47.05%     |
>
> These results demonstrate the significant value of our search framework over simple one-shot generation, showing that the iterative search and operator design are crucial for achieving SOTA performance. We will include these results as well as the o3 results for all search methods in the camera ready version of the paper.
>
> **Q2: Error analysis and code quality metrics**
>
> We appreciate the valuable suggestion. While the limited rebuttal timeframe prevents a full quantitative error analysis, our preliminary findings show the most common issues were ***data access KeyError exceptions**, **submission file formatting errors**, **third-party API KeyError exceptions e.g. lightgbm version issues**, and **execution timeouts** from inefficient data processing or computationally intensive models. We aim to provide a more systematic categorization of these errors and their frequencies in the camera-ready version.
>
> **Q3: Implementation of CROSSOVER operator**
> We apologize for this omission. The CROSSOVER operator combines elements from two parent solutions to create new candidate solutions by:
> Parent Selection: Two parent nodes are chosen using fitness-proportional selection based on validation scores
> Recombination: The LLM intelligently merges complementary strengths from both parents (e.g., feature engineering from Parent A + model architecture from Parent B) while maintaining code coherence
> We have added a subsection with implementation details and the prompt template into the appendix. The full technical specification will also be available in our open-source implementation.
>
> ```
> \paragraph{\textsc{Crossover} Operator} For two or more valid artifacts, the Crossover operator performs a merge of existing solutions. It specifically aims to create a solution drawing insights from the best aspects of its parents, while also taking into account the shared task context (see \Cref{subsec:aid_ops}) that informs and guides the merging process. For each crossover operation, the two parent nodes are sampled from the distribution of all nodes' fitness scores.
> ```
>
> See the template below.
>
> ```
>     # Introduction:
>     You are a Kaggle Grandmaster attending a high-stakes competition.
>     Your goal is to combine two previously developed solutions to further increase performance on the given task.
>     Carefully consider the task description, the two provided solutions, the available data, and compute resources.
>     You need to devise a plan to merge or integrate these solutions and then implement it.
>
>     # TASK DESCRIPTION
>     ```
>     {{task_desc}}
>     ```
>
>     {% if memory %}
>     # PREVIOUSLY EXPLORED CROSSOVER/COMBINATION IDEAS
>     ```markdown
>     {{memory}}
>     ```
>     {% endif %}
>
>     # PREVIOUS SOLUTION 1:
>     ## Code:
>     ```python
>     {{prev_code1}}
>     ```
>
>     # PREVIOUS SOLUTION 2:
>     ## Code:
>     ```python
>     {{prev_code2}}
>     ```
>
>     # INSTRUCTIONS:
>
>     Your main task is to:
>     1. Propose a **Crossover Plan** in natural language explaining how to combine Solution 1 and Solution 2.
>     2. Provide a **Python Code Implementation** of this plan.
>
>     Consider any previously explored ideas from the MEMORY section.
>     Brainstorm how the two provided solutions can be effectively combined and **WHY THIS CROSSOVER IS LIKELY TO BE EFFECTIVE** and increase performance for the given task, data, and compute resources.
>     Aim for a consistent evaluation method (e.g., 5-FOLD CROSS-VALIDATION, unless the task specifics dictate otherwise).
>
>     **CONSTRAINTS**:
>     - Be aware of the running time of the solution, it should complete within {{execution_timeout}}
>     - Prefer vectorized operations over Python loops when processing large datasets.
>     - Use `torch.optim.AdamW` (the recommended optimizer) instead of the deprecated `AdamW` from `transformers`.
>     - Replace the deprecated `early_stopping_rounds` argument in `lightgbm.train()` with the `lightgbm.early_stopping(stopping_rounds=…)` callback.
>     - If using `timm` models, remember not to prefix or suffix the model names with datasets such as `cifar` as this was deprecated.
>     - As much as possible, keep the stdout clean.
>
>     **DATA**: The data is already prepared and available in the read-only `./data` directory. You should not unzip any files.
>
>     **COMPUTE**: You have access to a Python environment with 1 NVIDIA H200 GPU(s) and 24 CPUs available, and the following packages installed: {{packages}}. If you need to, feel free to use additional libraries that fit the problem.
>
>     Start by making sure you understand the task, the provided solutions, the data and compute resources, and your proposed crossover idea. Then, generate a detailed internal implementation plan that will structure and guide you step-by-step through the implementation process. Make sure to reflect on the plan to ensure that the implementation is efficient, faithful to the crossover idea, and that all requirements (e.g., the evaluation score is printed, the submission file follows the correct format and is saved in the correct location, etc.) are satisfied.
>
>     **RESPONSE FORMAT FOR IMPLEMENTATION**:
>     Provide a **SINGLE** Markdown code block (wrapped in ```) containing a **SELF-CONTAINED** Python script that:
>     1. Implements the idea **END-TO-END**
>     2. **PRINTS THE 5-FOLD CROSS-VALIDATION** score of the evaluation metric
>     3. **SAVES THE TEST PREDICTIONS** in a `submission.csv` file in the current directory
>
>     Format the proposed solution as follows:
>
>     # Crossover Plan
>     <Your proposed crossover plan/strategy>
>
>     ```python
>     <the implementation of the crossover solution>
>     ```
> ```
>
> ### **Response to Minor Points**
>
> **Algorithmic novelty:** We agree that our work's key value lies in the systematic framework design and empirical investigation that leads to SOTA results. This systematic evaluation is enabled by our novel formalization, which also facilitates the research on the interplay of search and operators.
>
> **Line 184 clarification:** Thank you for spotting this ambiguity. We have corrected the sentence to read: "For all figures showing [OAI] results, we take AIDE O1 artifacts from the MLE-bench [4] GitHub repository, where O1 stands for O1-preview [32]."

---

> > ### Comment · Reviewer_DZn3 · 2025-08-09
> > **Re. author response**
> >
> > I thank the authors for their detailed responses to the reviewers. The authors have addressed my main questions satisfactorily. After reading the other reviews and comments, I did not see any new significant technical issues raised in the other reviews. Since the authors have agreed to include the error analysis in the final version, I remain positive about the paper and continue to recommend acceptance.

---

### Official Review · Reviewer_qCYX · 2025-07-02

**Clarity:** 4
**Significance:** 3
**Originality:** 3
**Rating:** 5
**Confidence:** 3

**Summary:**

The paper presents an analysis of AI research agents that solve the MLE benchmark, a set of Kaggle competitions. The authors discuss the important details of designing such agents, dividing the process into two major parts: high-level actions that are available to agents, and a search policy. The paper also investigates how the choice of proxy metric affects the agent's performance, and whether the gap between validation and test accuracies can be closed. Additionally, the authors propose to extend the action space of the current approach, which moderately boosts metrics.

**Questions:**

[1] In AIRAEVO explanation, it is said that there exists a Crossover operation. How is it implemented?

**Ethical Concerns:**

["NO or VERY MINOR ethics concerns only"]

**Final Justification:**

As my review states, I find this paper interesting and the results reliable. Thus, I am willing to accept this paper.

**Limitations:**

yes

**Paper Formatting Concerns:**

No major issues

**Quality:**

4

**Strengths And Weaknesses:**

**Quality**: This paper presents a comprehensive empirical analysis on AI research agents, with adequate experiment design that supports the claims. To strengthen the paper, an ablation for each additional operator might be included. The reported results seem reliable, with confidence intervals provided.

**Clarity**: The paper is easy to follow and clearly written, the difference between the approaches is highlighted, and each experiment is clearly presented.

**Significance and Originality**: The paper outlines major directions for improvement for future work, and therefore is beneficial for the scientific community and agentic research in particular.

---

> ### Author Rebuttal · Authors · 2025-07-31
>
> We sincerely thank Reviewer qCYX for their thorough evaluation and positive assessment of our work. We are particularly grateful for their recognition of our "comprehensive empirical analysis" with "adequate experiment design" and "reliable results," as well as the clarity of our presentation. Most importantly, we appreciate that the reviewer found our work to "outline major directions for improvement for future work" and be "beneficial for the scientific community and agentic research in particular."
>
>
> ### **Clarification on CROSSOVER Implementation**
>
> > **Q: In AIRA_EVO explanation, it is said that there exists a Crossover operation. How is it implemented?**
>
> Thank you for this important question, it is an oversight of ours that we did not properly explain. The CROSSOVER operator combines elements from two parent solutions to create a new candidate solution. Specifically, it takes two existing code artifacts and prompts the LLM to intelligently merge their complementary strengths—for example, combining the feature engineering approach from one solution with the model architecture from another.
>
> The implementation works as follows: when CROSSOVER is selected during evolutionary search, two parent nodes are chosen based on fitness-proportional selection by the selection policy. The operator is then provided the two parent codes with instructions to create a hybrid solution that leverages the best aspects of each parent while maintaining code coherence and functionality.
>
> **We will add the detailed explanation of the CROSSOVER implementation below to the Appendix of the revised paper with concrete examples to provide full technical details for reproducibility, as a companion to the open-source implementation.**
>
> ```
> \paragraph{\textsc{Crossover} Operator} For two or more valid artifacts, the Crossover operator performs a merge of existing solutions. It specifically aims to create a solution drawing insights from the best aspects of its parents, while also taking into account the shared task context (see \Cref{subsec:aid_ops}) that informs and guides the merging process. For each crossover operation, the two parent nodes are sampled from the distribution of all nodes' fitness scores.
> ```
>
> We also point the reviewer to the rebuttal for reviewer DZn3 to see the prompt template for the crossover operator.
>
> ### **Ablation Study for Individual Operators**
>
> We greatly appreciate the reviewer's suggestion to include "an ablation for each additional operator" to strengthen the paper. This would indeed provide valuable insights into the individual contributions of each operator in our AIRA set.
>
> While the current rebuttal timeline prevents us from conducting this specific ablation, we will aim to include these results in the camera-ready version of the paper.

---

> > ### Comment · Reviewer_qCYX · 2025-08-05
> >
> > I thank the authors for the rebuttal and I stand by my current score.

---

### Official Review · Reviewer_QQcd · 2025-07-03

**Clarity:** 3
**Significance:** 2
**Originality:** 2
**Rating:** 5
**Confidence:** 4

**Summary:**

This paper frames AI research tasks as a search problem and breaks down the design of AI research agents, such as AIDE, into two core components: search policies and operators. The authors build on this formulation to conduct a thorough ablation study, systematically evaluating the impact of different design choices for these components. The best configuration identified in the study achieves state-of-the-art results on the challenging MLE-Bench, improving the success rate for earning a Kaggle medal from 39.8% to 47% on MLE-Bench Lite. This result underscores the practical significance of their findings and highlights the value of carefully designing and evaluating agent components to advance automated machine learning research.

**Questions:**

- Figure 1a’s x-axis label “any medal” is confusing and unclear.
- Line 93 states that “fitness function F is defined for each node by the operator that generates or modifies it,” but Line 211 claims “each agent employs the same proxy-fitness function F.” These statements appear contradictory. Which is correct?
- Line 95 notes that “all operators are LLM-driven except for the MEMORY operator, which is defined by hand.” Does “LLM-driven” mean the operators use LLMs during execution or were generated by LLMs? Additionally, does "the MEMORY operator is defined by hand" imply that all other operators are LLM-generated (I guess not)?
- Line 105’s description of the MEMORY operator is unclear. If it does not involve an LLM, how is it implemented? If it does, this contradicts Line 95’s claim.
- Line 184 indicates that [DOJO] results use the R1 model, while [OAI] results use the O1 model. The use of different models introduces a confounder, making it difficult to draw conclusive comparisons.
- Line 194 suggests that a larger number of children (nc) requires more advanced code complexity. Why is this the case? Additionally, how can a DRAFT node have children, given its role in initiating solutions?
- Line 259 omits the exploration constant value c=0.5.

**Ethical Concerns:**

["NO or VERY MINOR ethics concerns only"]

**Final Justification:**

I think the authors resolve most of my concerns by providing more experimental results and detailed explanations to my questions, especially the newly added significance tests, which enhance the convincingness of the empirical conclusion, in my opinion.

I am still a bit concerned about the "validation-test" gap, but I believe that my other major concerns are addressed, and the remaining concern does not affect the major argument of the paper. I will increase my score to 5 and recommend the acceptance of the paper.

**Limitations:**

yes

**Quality:**

3

**Strengths And Weaknesses:**

### Strengths

1. The paper achieves a notable advancement by establishing a new state-of-the-art performance on MLE-Bench Lite, boosting the Kaggle medal success rate from 39.8% to 47.8%. This improvement highlights the practical impact of the proposed approach and sets a strong benchmark for future work in automated machine learning.

2. The authors provide an insightful perspective by decomposing the design of AI research agents into two key components: search policies and operators. Their comprehensive ablation studies offer valuable insights into how these components contribute to agent performance, paving the way for more systematic design of AI research agents.

3. The release of the codebase will be a significant strength, enabling other researchers to build upon this work and explore new designs for AI research agents. This commitment to open science enhances the paper’s impact and fosters further innovation in the field.

### Weaknesses

1. A primary concern is the absence of statistical significance tests to support the authors’ claims. For instance, in Line 263, the authors conclude that the search policy has minimal impact based on the small performance margin shown in Figure 4a. However, in Line 289, they argue that a slightly larger margin in Figure 3a suggests a meaningful interplay between search policy and operator design. Both conclusions are questionable, as the margins may fall within the error bars (as seen in Figure 3a) and lack statistical validation. Furthermore, the exploration of operator designs is limited, with only a brief comparison between OAI and DOJO, which is confounded by the use of different models, undermining the conclusiveness of the analysis. Including rigorous statistical tests and a broader exploration of operator designs would strengthen the paper’s findings.

2. The analysis in Section 5.3 raises concerns about the interpretation of the validation-test gap. The authors suggest that the model accessing such data represents an upper bound on performance. But both the validation and test sets are equally held out from the training data, with the only difference being whether they are included during the AI research agent search. This framing misrepresents the gap’s significance, as the test set is not inherently "ground-truth held out data". Additionally, the method described in Line 321 relies on test set data to report the best-performing scores, which is not feasible in real-world scenarios where test data is inaccessible during model development. I suggest removing this section from the paper.

---

> ### Author Rebuttal · Authors · 2025-07-31
>
> We appreciate Reviewer QQcd's thoughtful review and are pleased that they recognized our state-of-the-art performance on MLE-Bench Lite, our insightful decomposition of AI research agents into search policies and operators, and the value of our open-source commitment. We're also encouraged by their acknowledgment that our comprehensive ablation studies offer valuable insights into how these components contribute to agent performance. We address their concerns below.
> ## **Statistical Significance Testing and More Analysis**
> **We completely agree** that statistical tests would add more rigour to our conclusions. We have now performed Mann-Whitney U tests [1] to demonstrate the probability of improvement [2] of a method X over a method Y. When confidence intervals do not include 0.5, it indicates a statistically significant result [2]. We see that when comparing agents that utilize the AIDE operators, the probability of improvement is not statistically significant for any comparison besides MCTS c=0.5 where AIDE Greedy is better. Additionally, when comparing the AIRA operator agents, we see a statistically significant improvement with all AIRA agents except for the evolutionary search policy. **We will add these statistical significance results to the revised paper** to strengthen claims about performance differences between methods. These tests do not take the magnitude of improvement into account, only whether a method is more likely to perform better on a random task and random seed.
>
> **To address the comparison between OAI and DOJO, we conducted an experiment reproducing AIDE_greedy with o1-preview in AIRA-dojo**.
> Specifically, our implementation of AIDE, [DOJO] AIDE_Greedy o1-preview, improves the medal rate from 35% to 46% over the reported results, [OAI]  AIDE_Greedy o1-preview.
> This corresponds to state-of-the-art performance with a relative improvement of 30% in medal rate.
>
> While evaluating all agents with o1-preview would have been informative, we were only able to complete the AIDEgreedy experiment before the model was discontinued.
> However, **we additionally conducted experiments with OpenAI’s newest model in the series, o3, and observed similar results.**
>
> |Agent X|Agent Y|P(X > Y) Medal Rate|95% CI|
> |-|-|:-:|:-:|
> |AIRA_Evo o3|DOJO AIDE_Greedy o3|0.520|[0.491, 0.550]|
> |AIRA_MCTS o3|DOJO AIDE_Greedy o3|0.539|[0.511, 0.566]|
> |AIRA_Greedy o3|DOJO AIDE_Greedy o3|0.541|[0.511, 0.568]|
> |AIRA_Evo R1|DOJO AIDE_Greedy R1|0.516|[0.495, 0.535]|
> |AIRA_MCTS R1|DOJO AIDE_Greedy R1|0.536|[0.517, 0.557]|
> |AIRA_Greedy R1|DOJO AIDE_Greedy R1|0.528|[0.509, 0.548]|
> |DOJO AIDE_Greedy R1|OAI AIDE_Greedy o1-preview|0.524|[0.505, 0.543]|
> |DOJO AIDE_Greedy o1-preview|OAI AIDE_Greedy o1-preview|0.555|[0.530, 0.581]|
> |AIDE_EVO R1|AIDE_MCTS c=0.75 R1|0.519|[0.492, 0.546]|
> |AIDE_EVO R1|AIDE_MCTS c=0.0 R1|0.510|[0.484, 0.537]|
> |AIDE_EVO R1|DOJO AIDE_Greedy R1|0.493|[0.469, 0.516]|
> |AIDE_MCTS c=0.75 R1|AIDE_MCTS c=0.0 R1|0.491|[0.468, 0.514]|
> |AIDE_MCTS c=0.50 R1|DOJO AIDE_Greedy R1|0.476|[0.453, 0.497]|
> |AIDE_MCTS c=0.0 R1|DOJO AIDE_Greedy R1|0.483|[0.461, 0.503]|
>
> ### **Section 5.3 Validation-Test Gap Analysis**
> We appreciate your feedback regarding Section 5.3. We acknowledge that the motivation and implications of this section could be stated more clearly. We apologize if this did not come across as intended. We will clarify its importance and criticality for scaling research agents here, and we will update the writing to better reflect this for all readers.
>
> **As the reviewer points out, both the validation and the test set are sampled from the same distribution, and the main difference between the two is the fact that the agent uses the validation set during the search process. This is a critical distinction, which our experiments show leads to overfitting to the validation set.** Specifically, every time the agent queries the validation score and uses it to guide the search, it risks overfitting to the validation set due to finite sample effects. Because of that, the performance on the validation score seizes to be a good measure of performance, as suggested by Goodhart’s law.
> However, the test set, which is completely held-out, allows us to accurately estimate the actual expected error of the model, and study the gap between the perceived and actual expected error—the overfitting to the validation score during the search process.
>
> **How this gap evolves over time is critical for scaling AI research agents.
> The main point of this section is to empirically study the validation-test gap in the context of AI research agents.** We quantify the gap over time, show how it differs across different search policies, how it affects the effectiveness of different agents, and demonstrate overfitting. Unless researchers address this overfitting in the search process, scaling agents will be impossible. To the best of our knowledge, our study is the first to study this fundamental phenomenon in machine learning in the context of research agents.
>
> **Regarding Line 321 feasibility:** The section explores the maximum achievable (actual) test performance by selecting the top-k solutions based on the (perceived) validation scores. **This is useful and actionable information in many settings.** First, we could keep a held-out test and evaluate the top-k models on a held-out set and choose the best. Second, we could deploy several models in production, evaluate their performance in a small scale and choose the best (e.g., perform an A/B test). Finally, we could show several candidate solutions to machine learning experts that can to some extent account for overfitting, until we develop methods to do this automatically.
>
> This setting is even supported in Kaggle, where 2 submissions are allowed for the final private test set evaluation where the best submission is ultimately selected. This section demonstrates that after model development according to validation, if one were allowed k submissions and the best were selected, this would be the achievable score.
>
> **Nonetheless, we would like to emphasize again that the point of the section is not to show improved performance due to selection strategies. The section aims to study the relationship between validation and test scores within a search process and provide valuable insights that can assist future agentic development and could be readily actionable in some domains.**
>
> We hope that we clarified the value of this section to the review. We are happy to address any additional concerns that the reviewer might have. We will revise the section to make these points more clear to future readers
> ## **Minor Corrections**
> - **Figure 1a x-axis:** Agreed—we will remove the "any medal" label as the y-axis already specifies medal rate.
> - **Line 259 missing constant:** Thank you for catching this. We will add c=0.5 to the revised manuscript.
> ## **Clarifications**
> - **L93 vs L211 Fitness Function:** Sorry for the confusion—this is a good catch. The fitness function can be: (1) an operator that assigns values to nodes, (2) ground truth evaluation for verifiable domains, or (3) a byproduct of operator execution (e.g., 5-fold CV from DRAFT). For Kaggle problems with hidden test sets, each DRAFT/IMPROVE solution performs 5-fold CV, which is extracted as the fitness function. **We will clarify this distinction in the description.**
> - **L95 & L105 "LLM-driven" vs "defined by hand":** "LLM-driven" means the operator's execution involves an LLM call in some way. The MEMORY operator's logic (retrieving previous node information) is hand-coded and deterministic, requiring no LLM call. It outputs plans and validation results in structured format in which each node’s information appended on top of each other separated by a line divider.
> - **L184 Model Confounding (R1 vs O1 (preview)):** We acknowledge this limitation. Different models were used to ensure fair comparison with the original AIDE baseline. **To address this concern, we have included o1-preview results in DOJO and ran o3 on all search/operator combinations**—these results will be in the revised paper, showing infrastructure impact alone is significant. Whilst, ideally we would run o1-preview with all the search methods, the model was deprecated from OpenAI’s services before we could run the experiments.
> - **L194 Complexity and DRAFT Children:** We will clarify this description. nc defines how many children a node currently has. Our complexity rule ensures siblings created sequentially become increasingly complex—ensuring simple solutions are created first and complex solutions later. This is not stating that having more children means the node **needs** to be more complex. This is just a heuristic way to control the complexity of the planned solutions throughout the search tree.
> - **DRAFT nodes do have children:** They are initial nodes from the blank root node, with the IMPROVE operators creating their children. E.g. If you called IMPROVE on the same draft node 5 times, it would have 5 separate children stemming from it.
> ## **Summary**
> We believe the clarifications concerning the overfitting experiments and the additions (statistical tests, additional experiments, and improved descriptions) will significantly strengthen the paper and its core contributions. We appreciate the reviewer's constructive feedback in helping us improve the work.
>
> We would appreciate the reviewer’s consideration of increasing their score, as we believe we have adequately addressed the main limitations they identified. If the review has additional concerns about our responses, we kindly ask them to let us know.
> - *[1] Mann, Henry B., and Donald R. Whitney. "On a test of whether one of two random variables is stochastically larger than the other." The annals of mathematical statistics (1947): 50-60.*
> - *[2] Agarwal, Rishabh, et al. "Deep reinforcement learning at the edge of the statistical precipice." Advances in neural information processing systems 34 (2021): 29304-29320.*

---

> > ### Comment · Reviewer_QQcd · 2025-08-05
> >
> > I thank the authors for providing more experimental results and detailed explanations to my questions. I think they resolve most of my concerns, especially the newly added significance tests, which enhance the convincingness of the empirical conclusion, in my opinion.
> >
> > However, I am still a bit concerned about the "validation-test" gap. I think we both agree that both the validation and the test set are sampled from the same distribution. But I don't agree that this "gap" can be empirically estimated by the proposed method. For other similar cases, such as proxy evaluation and true evaluation signal, I agree that the proposed method can measure a meaningful gap for the quality of the proxy evaluation signal. But in the reported case, I don't think the "upper bound" is convincingly meaningful, which only shows a number that we tested on a datasplit that was already exposed to the "training" of the model.
> >
> > That being said, I believe that my other major concerns are addressed, and the remaining concern does not affect the major argument of the paper. I will increase my score to 5 and recommend the acceptance of the paper.

---

### Official Review · Reviewer_z7Ae · 2025-07-07

**Clarity:** 3
**Significance:** 1
**Originality:** 2
**Rating:** 3
**Confidence:** 4

**Summary:**

This paper provides a new perspective where the design of agents is 1) Search policy, 2) Operator that modifies the current solution.
A formalization is provided where AIDE is re-interpreted under this new lens.
They also provided AI Research Agent (AIRA) Dojo, a software/framework that allows people to implement new operators and search policies.
The paper also found a generalization gap.

**Questions:**

N/A

**Ethical Concerns:**

["NO or VERY MINOR ethics concerns only"]

**Final Justification:**

My score is final -- which is an indication of my preference. Since all the other reviewers believe this is a good paper, as research is a community-driven effort, I won't object to sharing this paper with the broader public to inspire future work.

**Limitations:**

Yes

**Quality:**

3

**Strengths And Weaknesses:**

Strengths:
1. The paper is indeed well-written and easy to follow.
2. The re-framing is a good attempt to provide insight into this field.

Weaknesses:

I believe this is a borderline paper. I can either vote to accept or reject, but will not take a strong stance to vote either way. The paper is well-written, and the experiments are solid. However, the paper puts a lot of emphasis on the **reframing** of AIDE and agents. Then, whether I personally is convinced by the **reframing** is the key to whether I should vote to accept or reject this paper.

I do not find this reframing convincing. My biggest confusion is -- for an agent that has access to tools (such as terminal/internet), what is the search policy and operator? The paper seems to want to merge two separate conceptual models: agent and inference-time search. I think they are different. The authors seem to think they are the same. I'm open to having a longer discussion with the authors during the rebuttal.

The paper can also use better citations, such as [1] and [2]. The overfitting phenomenon was also discovered by Anthropic last year [3], but proper citation was not provided.

[1] https://deepmind.google/discover/blog/alphaevolve-a-gemini-powered-coding-agent-for-designing-advanced-algorithms/ and the paper: Novikov, A., Vũ, N., Eisenberger, M., Dupont, E., Huang, P. S., Wagner, A. Z., ... & Balog, M. (2025). AlphaEvolve: A coding agent for scientific and algorithmic discovery. arXiv preprint arXiv:2506.13131.

[2] Pan, Jiayi, Xiuyu Li, Long Lian, Charlie Snell, Yifei Zhou, Adam Yala, Trevor Darrell, Kurt Keutzer, and Alane Suhr. "Learning adaptive parallel reasoning with language models." arXiv preprint arXiv:2504.15466 (2025).

[3] Pan, Jane, He He, Samuel R. Bowman, and Shi Feng. "Spontaneous Reward Hacking in Iterative Self-Refinement." arXiv preprint arXiv:2407.04549 (2024).

---

> ### Author Rebuttal · Authors · 2025-07-31
>
> We thank Reviewer z7Ae for describing the paper as “well-written and easy to follow,” the experiments as “solid,” and the reframing as a “good attempt to provide insight into this field.”
>
> The reviewer's concerns center on the relation between research agents and inference-time search, and how ReAct-like LLM-driven agents with access to tools fit into the framework. We acknowledge that understanding how these types of agents fit within the framework is very useful and important for a reader. Given that the reviewer has confirmed the technical soundness of our experiments, we will focus on addressing the conceptual concerns and clarifying the practical significance of our work.
>
> > **Reviewer's Concern:** "The paper seems to want to merge two separate conceptual models: agent and inference-time search."
>
>
> Research agents aim to automate the scientific research process by **taking actions** that allow them to effectively **(re)search** the possible space of solutions, by iteratively coming up with and testing hypotheses that best **inform the subsequent actions**, until they reach a satisfactory solution (or run out of time). Conceptualizing this process as search is natural, and allows for a systematic study of the exploration-exploitation tradeoffs inherent to search and extensively studied in the literature (see next section). Our work formalizes this perspective by modeling research agents as graph-based search algorithms. Crucially, this abstraction does not impose constraints on the implementation of the agent and the agents’ action space (see next section)—a node is a conceptual object that can span many actions and an operator can be another fully-fledged agent that performs many steps (see Complex operators in Section 7.1). In fact, in our implementation, nodes are composed of multiple operator calls (e.g., the core operator such as debug or improve and the analysis operator that extracts information from the execution output).
>
> We do not use the term “inference-time search” in the paper, and if the reviewer uses it to refer to "performing search" internally by generating more thinking tokens, then we are not arguing that these are the same. In fact, this is not agentic, and is strictly complementary: we are relying on reasoning models in our experiments, and we are agnostic to how they use the "thinking tokens". We hope that this discussion clarifies the relation between research agents and search in the context of the proposed formalization.
>
>
> Does the reviewer have any additional questions regarding this concern, or believe that a reader would benefit from including some clarification in the paper?
>
> > **Reviewer’s Concern: "For an agent that has access to tools (such as terminal/internet), what is the search policy and operator?"**
>
> Consider a standard ReAct-like LLM-driven agent with access to tools solving a coding problem on MLE-Bench. There are many ways to conceptualize such agents in our framework, depending on the definition of the artifacts set that define the nodes and the operator. Here we provide **an example instantiation of a ReAct agent in our framework**:
>
> **Nodes:** (Partial) Solution. We could keep this as in the paper’s experiments where each node represents a candidate solution (code file) and some meta-data.
>
> **Operators (O):** A single operator that generates the next solution (based on all of the previous information in the context; note that the context will grow relatively quickly) wrapping the logic of the ReAct agent. The agent could take several actions (note that we make no assumption about the low-level action space)—observe the current state, reason about the next action, execute a tool (terminal/internet), and update the state—until it generates the next candidate solution.
>
> **Search Policy (π):** "Linear search" over N steps. At each step, the search policy selects the last node for expansion, or decides to terminate the search. This is a trivial search policy. **This type of agent does not use the search policy as a sophisticated means of modulating the inherent exploration-exploitation tradeoff.** There is a single operator, so the operator policy is trivial in this case.
>
> **Key insight:** Even "simple" tool-using research agents implement a search: a systematic exploration of the solution space through a sequence of actions toward finding a solution to the problem. The only difference is that they use “simple linear search" rather than more sophisticated search policy like MCTS.
>
> **Why is this significant?**
>
> **The proposed formalism enables systematic analysis. Our experimental results expose bottlenecks in research agent design and identify directions for improvement for future work.**
>
> Specifically, the framework allows us to describe the agent’s behavior in terms of fundamental components that shape its behavior: 1) the **search policy:** deciding where resources will be allocated; 2) the **operator set**, the high-level functions that take in existing artifacts and produce new ones, and the **operator policy**; 3) the evaluation mechanism used to score artifacts and guide the search towards promising directions. This allows us to compare agents that are seemingly not comparable and disentangle the impact of specific design choices. In particular:
> - **AIDE** = greedy search with specific draft, improve, and debug operators
> - **AlphaEvolve** = evolutionary search with specific mutation and crossover operators
> - **ReAct agents** = linear search with a specific code-generation operator
>
> In our work, we focus on the previous state-of-the-art AIDE and LLM-based operators: changing the search policy while keeping the operators fixed, and vice-versa. **Our comprehensive analysis exposes a complex interplay between the search components**. For example, components can act as bottlenecks to other components (e.g., operators or overfitting can bottleneck potential gains from search strategies), the overfitting manifests differently for different search algorithms, and the exploration-exploitation tradeoff can be modulated by the operators or/and by the search policy.
>
> **We appreciate the reviewer's offer for extended discussion.**
>
> **Missing Prior Work Citations**
>
> We thank the reviewer for flagging these missing citations, and have incorporated the suggested citations into our revised manuscript:
> - **[1] AlphaEvolve & [2] Pan et al. (Adaptive Parallel Reasoning):** These works illustrate the framework's applicability:
>   - **AlphaEvolve** = evolutionary search policy + sophisticated code-mutation operators *(Submitted on 16 Jun 2025)*
>   - **Pan et al.** = advanced parallel search policy with dynamic resource allocation *(Submitted on 21 Apr 2025)*
>
> - **[3] Pan et al. (Reward Hacking):** Thank you for making us aware of this paper. We have added this reference to our discussion of the generalization gap. The phenomenon they term "Spontaneous Reward Hacking" relates closely to the overfitting we observe. Our MLE-Bench results provide additional empirical evidence of this phenomenon in automated AI research, confirming and extending their insights to this domain as well as seeing the effect of different search policies on this overfitting.
>
>
> Overall, we contend that this work provides **conceptual** (the formalism) and **practical** (AIRA-Dojo) tools for advancing AI research automation. The insights from the paper support the value of these tools.
>
> The reviewer’s “borderline” rating and “poor” significance are centered around the conceptual framing and generalization. The reviewer acknowledges our solid experiments and clear writing, and we hope that the conceptual clarifications above help to address the generalization concern and contextualize the significance of our contributions. If that is the case, we respectfully encourage the reviewer to consider revising their assessment.
>
> We welcome further dialogue of any remaining concerns about the conceptual framing, or how agent architectures fit into the framework. We sincerely thank the reviewer for helping us make the paper better.

---

> > ### Comment · Reviewer_z7Ae · 2025-08-05
> >
> > The paper seems to be warmly received by the community -- it is not my preference, but I won't object to accepting this paper.

---

> > > ### Author Response · Authors · 2025-08-05
> > >
> > > Thank you for the helpful feedback, which has contributed to improving the paper. Are there any remaining concerns or questions that we could address—either through further discussion or additional results—to help clarify the work and improve the reviewer’s rating?

---

### Comment · Area_Chair_ZSmz · 2025-08-04

I would like to sincerely thank all reviewers for their thoughtful evaluations and the authors for their detailed rebuttal.
As we move into the discussion phase, I kindly remind reviewers to engage in further dialogue and respond to the rebuttal responses. Your continued input will be invaluable in helping me make a well-informed final decision. Thank you again for your contributions.

---

### Comment · Area_Chair_ZSmz · 2025-08-08

Thank you again to all the reviewers who have engaged in the rebuttal phase for discussion!

@Reviewer DZn3, I note that you haven't replied to author's rebuttal. Please consider submitting your response ASAP as we are approaching to the discussion phase deadline. Please note that submitting Mandatory Acknowledgement without a response is not the right way for rebuttal discussion.
Thank you!

For all other reviewers, please confirm that you have finalized your comments/ratings. Thank you very much.

---

### Decision · Program_Chairs · 2025-09-17

**Decision:**

Accept (spotlight)

**Comment:**

(a) Summary of claims and findings
This paper studies AI research agents in the context of MLE-Bench, framing them as search policies applied over candidate solutions with operators that iteratively modify code. The authors introduce AIRA (a formal framework for describing agents in terms of search policies and operators) and AIRA-Dojo (an open-source environment for experimenting with different agent designs). They perform extensive ablations across search strategies (greedy, evolutionary, MCTS) and operator sets, and demonstrate that the interplay between these components is critical. Their best configuration achieves a new state-of-the-art on MLE-Bench Lite, raising Kaggle medal success from 39.8% to 47%. The paper also highlights a validation–test gap that reflects overfitting during agentic search.

(b) Strengths
- Clear and systematic framework for analyzing research agents, with a useful decomposition into search policies and operators.
- Strong empirical contribution: extensive experiments and ablations that expose how search strategy and operator design interact.
- Achieves a new state-of-the-art on MLE-Bench Lite, a challenging and competitive benchmark.
- Provides open-source infrastructure (AIRA-Dojo), which will benefit the community.
- Identifies and analyzes generalization gaps and overfitting phenomena, offering insights relevant beyond this benchmark.
- Writing is generally clear and accessible, and the results are presented in a reproducible way.

(c) Weaknesses
- Algorithmic novelty is somewhat limited, as the work builds heavily on AIDE and uses standard planning algorithms. The main contribution lies more in empirical systematization than in new methods.
- Some conceptual framing (e.g., unifying agents and inference-time search) was not fully convincing to one reviewer.
- Evaluation does not include direct comparisons to the latest reasoning/code LLMs as zero-shot baselines, though rebuttal results partly addressed this.
- The validation–test gap section remains somewhat debated in terms of interpretation.

(d) Reasons for decision
The strengths outweigh the weaknesses. The paper is technically solid, makes a clear empirical contribution, and introduces infrastructure that will likely spur further work. Although the novelty lies more in systematization and analysis than new algorithms, the comprehensive study, new SOTA results, and open-source release together represent a meaningful step forward for the community. The additional insights about overfitting and generalization in agentic search add to its significance. These merits place the paper above a standard poster; it makes a strong case for spotlight as a well-executed empirical and infrastructural contribution with visible community impact.

(e) Discussion and rebuttal period
Reviewers initially disagreed on the conceptual framing and significance. The rebuttal clarified how tool-using agents fit into the proposed framework, added missing citations, and provided new experiments (including statistical significance tests and SOTA LLM baselines). These efforts addressed most concerns, with reviewers updating their scores upward or softening objections. Some residual reservations about framing and validation–test interpretation remain, but overall consensus shifted to positive, with multiple reviewers recommending acceptance. Given the strong experimental depth, open-source contribution, and new SOTA benchmark results, I recommend acceptance as spotlight.